# Short activation domains control chromatin association of transcription factors

**Vinson B Fan[1], Abrar A Abidi[1], Thomas GW Graham[1,2], Xavier Darzacq[1], Max V Staller[1,3,4]\***

[1]Department of Molecular and Cell Biology, University of California, Berkeley, United States; [2]Thomas C. Jenkins Department of Biophysics, Johns Hopkins University, Baltimore, United States; [3]Biohub, San Francisco, United States; [4]Center for Computational Biology, University of California, Berkeley, United States

**Abstract** Transcription factors regulate gene expression with DNA-binding domains (DBDs) and activation domains. Despite evidence to the contrary, DBDs are often assumed to be the primary mediators of transcription factor (TF) interactions with DNA and chromatin. Here, we used fast single-molecule tracking of transcription factors in living cells to show that short activation domains can control the fraction of molecules bound to chromatin. Stronger activation domains have higher bound fractions and longer residence times on chromatin. Furthermore, mutations that increase activation domain strength also increase chromatin binding. This trend was consistent in four different activation domains and their mutants. This effect further held for activation domains appended to three different structural classes of DBDs. Stronger activation domains with high chromatin-bound fractions also exhibited increased binding to the p300 coactivator in proximity-assisted photoactivation experiments. Genome-wide measurements indicate these activation domains primarily control the occupancy of binding rather than the genomic location. Taken together, these results demonstrate that very short activation domains play a major role in tethering transcription factors to chromatin.

**\*For correspondence:**
Mstaller@berkeley.edu

[†]Co-first author.

## Editor's evaluation

This useful study presents evidence that the duration and/or the frequency with which transcription factors interact with chromatin/DNA in living cells is influenced by the transactivation domains of transcription factors. The methods used are solid, combining imaging and genomics experiments. The work will be of interest to molecular biologists and biochemists, working in the transcriptional regulation field.

## Introduction

Transcription factors (TFs) contain DNA-binding domains (DBDs) that bind cognate DNA motifs and effector domains that regulate transcription (*Brent and Ptashne, 1985*; *Keegan et al., 1986*; *Ptashne, 1988*; *Struhl, 1989*; *Struhl, 1987*). DBDs are ordered, conserved, and readily predicted from protein sequences (*El-Gebali et al., 2019*; *Finn et al., 2016*). Decades of biochemistry have shown that DBDs are sufficient to bind cognate DNA motifs in vitro (*Riley et al., 2014*; *Stormo, 2013*). As a result, for most human TFs, only the DBD was used to define the position weight matrices that describe DNA-binding specificity (*Badis et al., 2009*; *Hume et al., 2015*; *Jolma et al., 2013*; *Latchman, 2008*; *Noyes et al., 2008*). Outside the DBD, TFs are predominantly composed of long intrinsically

**eLife digest** The human body is made up of about 200 different types of cells. Although all these cells contain the same genetic instructions in their DNA, each type performs specific functions. This is possible because different cells activate different sets of genes; for example, liver cells switch on genes specific to the liver, while blood cells turn on blood-specific genes.

Proteins known as transcription factors bind to DNA and activate specific genes, and they play a critical role in generating gene expression patterns that are unique to each cell type. Transcription factors are thought to be modular with two functional elements: DNA-binding domains that attach to genes, and activation domains that recruit the transcriptional machinery to switch on genes.

DNA-binding domains have traditionally been considered solely responsible for genome interaction. However, recent research indicates that short segments within the activation domain can also influence the transcription factors' affinity for DNA binding. To examine this, Fan et al. tracked individual transcription factor molecules in living cells and quantified the proportion bound to the genome.

The results confirmed that short activation domains play a major role in controlling the fraction of transcription factor molecules that bind to the genome. Strong activation domains – which interact robustly with coactivators or other components of the transcriptional machinery – increased the fraction of transcription-factor molecules bound to the genome. This effect was also achieved experimentally by strengthening otherwise regular activation domains through targeted mutation.

The study further demonstrated that activation domains tether transcription factors to the genome by binding to a coactivator already associated with chromatin, a protein that packages DNA into compact parcels. This binding may indirectly anchor the transcription factor to DNA even before its own DNA-binding domain engages.

Understanding how transcription factors bind the genome is essential for elucidating gene regulation. Traditional models assume that transcription factors first bind DNA through their DNA-binding domains and then use their activation domains to recruit the transcriptional machinery. The study of Fan et al. supports a model in which activation domains can instead bind coactivators that are already bound to DNA, tethering transcription factors indirectly to chromatin. These insights may ultimately deepen our understanding of gene regulation in development and disease.

disordered regions (IDRs) that contain short repression domains, which bind corepressor complexes, and activation domains, which bind coactivators (*Hahn and Young, 2011*; *Liu et al., 2006*; *Már et al., 2023*; *Kumar et al., 2023*; *Soto et al., 2022*). Activation domain function is separable from DBD function, as demonstrated by cut-and-paste experiments where activation domains retained function when attached to multiple DBDs (*Hope et al., 1988*; *Hope and Struhl, 1986*; *Ma and Ptashne, 1987*; *Sadowski et al., 1988*; *Sigler, 1988*; *Struhl, 1993*, *Struhl, 1987*).

Both DBDs and activation domains contribute to DNA binding and chromatin binding. DNA binding results from direct protein-DNA interaction is well established for DBDs in vitro and is measured in vivo with single-molecule footprinting (*Doughty et al., 2024*; *Stormo, 2013*). Chromatin binding includes DNA binding but can also result from protein-protein interactions with histones, other TFs, corepressors, or coactivators that can indirectly tether a TF to DNA (*Spitz and Furlong, 2012*). Nearly all cell-based assays of TF genome localization (e.g. ChIP-seq or CUT&RUN) measure the combination of DNA binding and chromatin binding. Some activation domains interfere with DNA binding (*Golemis and Brent, 1992*). In vitro, interactions between activation domains and DBDs can increase DNA-binding specificity (*Krois et al., 2018*; *Liu et al., 2008*; *Ning et al., 2022*; *Sun et al., 2021*). IDRs can increase nonspecific binding (*Baughman et al., 2022*), regulate DNA binding via posttranslational modifications (*Pufall et al., 2005*), or allosterically regulate DNA binding (*Li et al., 2017*). There is long-standing evidence that the DBD does not confer all genomic targeting information because in ChIP-seq experiments, 30–70% of peaks lack a motif for the query TF (*Harrison et al., 2011*; *Jana et al., 2021*; *Kvon et al., 2012*; *Spitz and Furlong, 2012*; *Teytelman et al., 2013*). There is also low agreement between the promoters a TF binds and the genes it regulates (*Mahendrawada et al., 2025*). Finally, there is extensive literature demonstrating how long IDRs—and not DBDs—control TF genomic localization (*Brodsky et al., 2020*; *Gera et al., 2022*; *Hurieva et al., 2024*; *Jonas et al.,*

*2023*; *Kumar et al., 2023*; *Mindel et al., 2024a*; *Mindel et al., 2024b*). These studies show that long IDRs are necessary and sufficient to localize TFs to target promoter genes; minimal activation domains were reported not to contribute (*Brodsky et al., 2020*). Although the original data could not distinguish between direct DNA binding and chromatin binding via protein-protein interactions with DNA-bound factors (reviewed in *Staller, 2022a*), recent studies suggest an IDR can bind DNA directly (*Strugo et al., 2025*). This genomic targeting by IDRs has also been seen in human cells (*Goldman et al., 2023*). These findings can explain why removing the DBD can increase the number of genomic binding sites detected by ChIP-seq (*Chen et al., 2014*), have little effect on genome binding (*Cowling and Cole, 2007*), or complement null mutants (*Copeland et al., 1996*). We have shown that long IDRs control the fraction of TF molecules bound to chromatin for factors in the HIF family (*Chen et al., 2022*). Other single-molecule imaging studies reported that IDRs impact nuclear diffusion but not genome binding (*Callegari et al., 2019*; *Mazzocca et al., 2023*). A single-molecule footprinting assay found that a strong activation domain can increase TF occupancy (*Doughty et al., 2024*). Together, these studies demonstrate how long IDRs or activation domains modulate chromatin binding.

In this study, we asked whether short activation domains within an IDR are sufficient to control chromatin association. While both DBDs and activation domains contribute to DNA binding and chromatin binding, the relative magnitude of these contributions cannot be predicted and must be measured. Activation domains recruit remodelers, like BAF, which move nucleosomes (*Kadoch and Crabtree, 2015*), coactivators, like p300/CBP, which modify chromatin (*DelRosso et al., 2024*; *Dyson and Wright, 2016*), and general transcription components like the Mediator complex (*Currie et al., 2017*; *Henley et al., 2020*; *Lauberth et al., 2013*; *Malik and Roeder, 2010*; *Tang et al., 2013*) that recruits RNA polymerase II (*Alerasool et al., 2022*; *Gill et al., 1994*). Although mechanistic understanding of activation domains still lags far behind DBDs, recent screens have cataloged many activation domains (*DelRosso et al., 2023*; *Erijman et al., 2020*; *Morffy et al., 2024*; *Sanborn et al., 2021*) and shown how the strength of acidic activation domains, the oldest and largest class (*Sigler, 1988*), depends on a balance of acidic, aromatic, and leucine residues (*DelRosso et al., 2023*; *Staller et al., 2018*; *Staller et al., 2022b*; *Udupa et al., 2024*). For consistency, we define activation domain strength as the fluorescence of a GFP reporter construct driven by a heterologous promoter.

Using single-molecule tracking (SMT) in live cells, we find that activation domains as short as 39–60 amino acids strongly affect the fraction of synthetic TF molecules bound to chromatin. TFs with a higher bound fraction have increased residence time on chromatin as measured by fluorescence recovery after photobleaching (FRAP). The chromatin-bound fraction and residence time of a series of activation domain mutants correlated with their ability to activate transcription. This effect was observed with activation domains attached to three structurally diverse DBDs, indicating that activation domains can control TF association with chromatin independent of DBD identity. Using pharmacological perturbations and proximity-assisted photoactivation (PAPA) experiments (*Graham et al., 2022*), we demonstrate this chromatin association results from activation domains binding to coactivator complexes. CUT&RUN experiments revealed all the synthetic TFs bound to overlapping sets of ATAC-seq accessible loci, suggesting that activation domains are binding to coactivators already engaged with chromatin. Stronger activation domains showed more binding to the same loci. These results suggest a key aspect of activation domain function could derive from increasing the ability of a TF to interact with chromatin.

## Results

### SMT of a synthetic TF

To test whether short activation domains are sufficient to control chromatin binding of TFs, we performed SMT on a synthetic factor. This synthetic TF was developed for high-throughput reporter studies of activation domain function (*Staller et al., 2022b*) and contains an estrogen-binding domain for inducible activation and six synthetic C2H2 zinc fingers (*Park et al., 2019*) engineered to bind to a DNA motif that is not found in the human genome. To facilitate SMT, we added a HaloTag for labeling with photostable synthetic dyes (*Grimm et al., 2021*; *Figure 1A and B*). At the N-terminus of this synthetic TF, we fused candidate activation domains and their rationally designed mutants. This synthetic TF is expressed from the weak L40 promoter and has low protein stability (*Staller et al., 2022b*), resulting in low steady-state expression levels. We engineered the cognate reporter into

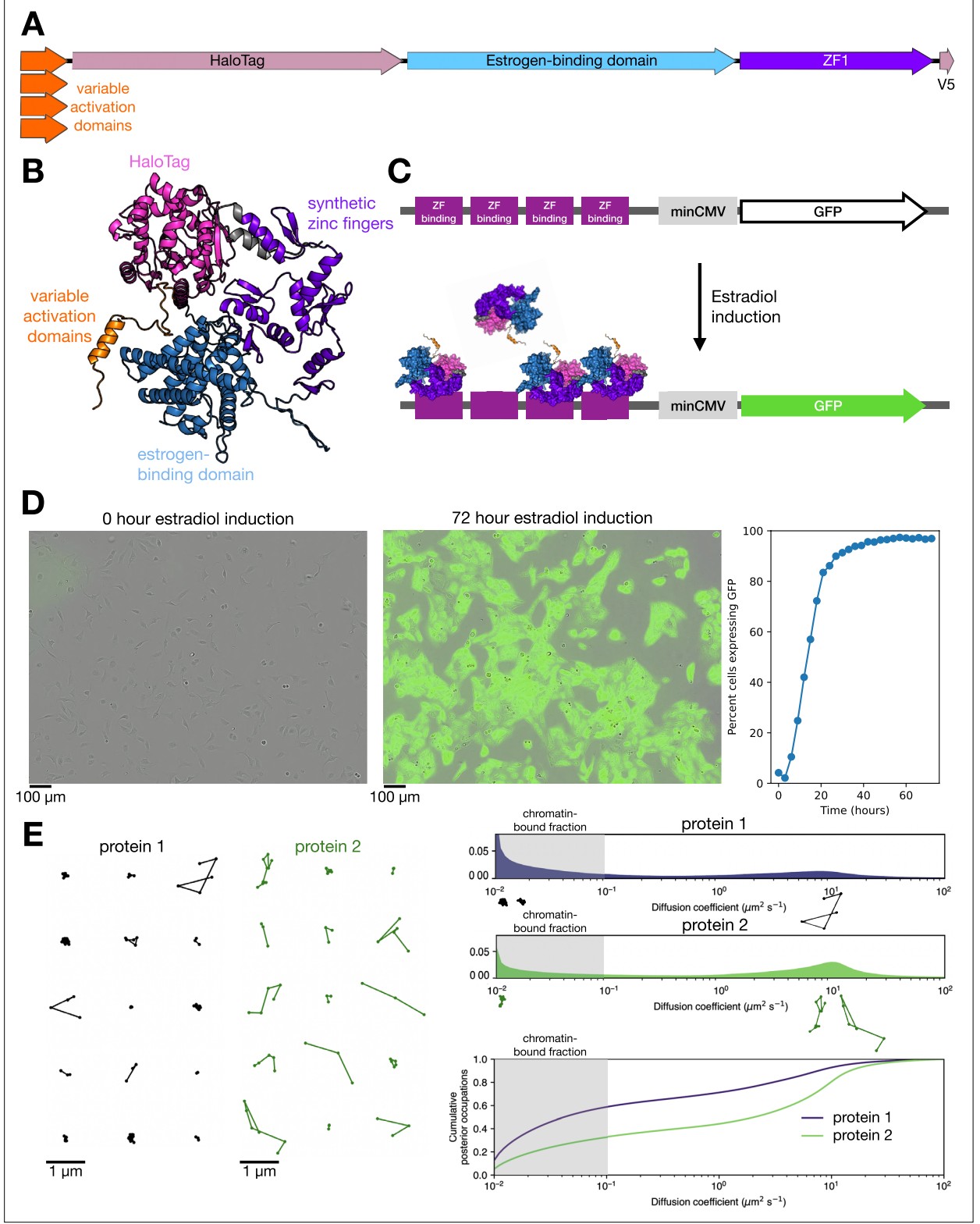

**Figure 1.** A synthetic transcription factor (TF) for quantifying chromatin association and reporter activity. (**A**) Schematic of the synthetic TF used in this study: activation domains (39–60 amino acids) are fused to HaloTag, an estrogen-binding domain, six synthetic C2H2 zinc fingers (ZF1), and a V5 epitope tag. Black line indicates linkers between domains. (**B**) AlphaFold2 model of the synthetic TF. (**C**) Schematic of the synthetic reporter locus (not to scale) installed at AAVS1. Four zinc-finger binding sites are engineered upstream of a minimal promoter, minCMV. These elements are upstream of GFP, which is produced in response to estradiol induction. (**D**) Overlay of GFP and phase-contrast images of reporter-bearing cells expressing VP16-synthetic TF

*Figure 1 continued on next page*

*Figure 1 continued*

at 0 hr (left) and 72 hr (right) after estradiol induction. Quantification of the percentage of GFP-positive cells over time from live imaging is shown in the right plot. (**E**) Left: Trajectories extracted from single-molecule tracking (SMT) movies from cells expressing two example proteins. Thousands of these trajectories are pooled to generate inferred diffusion spectra, right. Lower panel indicates the cumulative distribution function of the diffusion spectra. We consider the shaded gray region (D<0.1 µm²/s) the chromatin-associated fraction of a protein.

the AAVS1 locus of U2OS cells and derived a monoclonal line to assay transcriptional activity by GFP expression (*Figure 1C*, Materials and methods). This reporter locus is the only perfect motif match for the synthetic DBD in the genome. We stably integrated each synthetic TF into this cell line and verified that estradiol induction robustly induces GFP expression (*Figure 1D*).

We performed fast SMT on these synthetic TFs using highly inclined and laminated optical sheet (HILO) illumination at fast (~7.5 ms) frame rates (*Figure 1E*, *Video 1*), localized and tracked single molecules, and a published Bayesian mixture model to infer the distribution of their diffusion coefficients (*Heckert et al., 2022*). We define proteins with a diffusion coefficient D<0.1 µm²/s as immobile and assume immobile molecules are bound to chromatin (*Chen et al., 2022*; *Ferrie et al., 2024*). Importantly, this chromatin association does not necessarily indicate direct binding to DNA but could include interactions with factors directly bound to DNA. Molecules with higher diffusion coefficients are untethered and are likely freely diffusing or diffusing in complex with other factors (*Ferrie et al., 2022*; *Staller, 2022a*). In this work, we focus on the fraction of immobile molecules bound to chromatin.

## Short activation domains control the fraction of TF molecules bound to chromatin

Given the evidence that long IDRs can control the fraction of molecules bound to chromatin, we tested the hypothesis that minimal activation domains can also contribute to this phenotype. In previous examples, the IDRs that controlled the bound fraction were the majority of the protein (*Chen et al., 2022*). Here, our synthetic TF is ~800 AAs and the activation domain is 39–60 residues on the N-terminus.

Despite comprising only 5–7% of the total polypeptide sequence, these short activation domains dominated the chromatin-bound fraction of our synthetic TF. Three acidic activation domains derived from VP16, CITED2, and HIF1α each yielded different bound fractions and diffusion spectra (*Figure 2A*). These differences were also apparent in jump-length histograms and generally agreed with parameters modeled from Spot-On (*Hansen et al., 2018*; *Figure 2—figure supplement 1*), independently validating this result. This bound fraction was positively correlated with GFP expression of the reporter locus measured by flow cytometry (*Figure 2B and C*), suggesting a link between the strength of activation domains and their propensity to associate with chromatin (*Figure 2D*). The synthetic TF without an activation domain (hereafter the empty TF) also had a low (but not the lowest) fraction of molecules bound to chromatin.

As previously reported (*Driouchi et al., 2025*; *Walther et al., 2024*), we find substantial cell-to-cell variability when making SMT measurements in these cell lines (*Figure 2—figure supplement 2*), which necessitated large datasets with many cells and trajectories for reporting reproducible diffusion spectra (Materials and methods; *Supplementary file 2*). The difference in bound fractions also held when these synthetic TFs were transfected into cells without reporter loci (*Figure 2—figure supplement 3*), implying that the majority of immobile molecules bound nonspecifically to genomic sites without a match to the cognate motif.

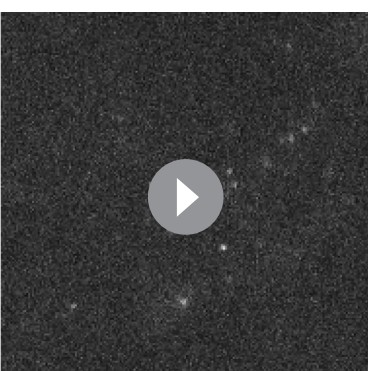

**Video 1.** Exemplary single-molecule tracking (SMT) movie of a cell expressing VP16-synthetic transcription factor (TF). SMT movie shown in real time of a cell expressing VP16-synthetic TF. Trajectories overlaid and colored according to their mean jump length, where brighter-colored spots belong to faster-moving trajectories. Initially, frames are dense with detections and are rejected for analysis. Detections outside a curated nuclear mask are also rejected.

https://elifesciences.org/articles/105776/figures#video1

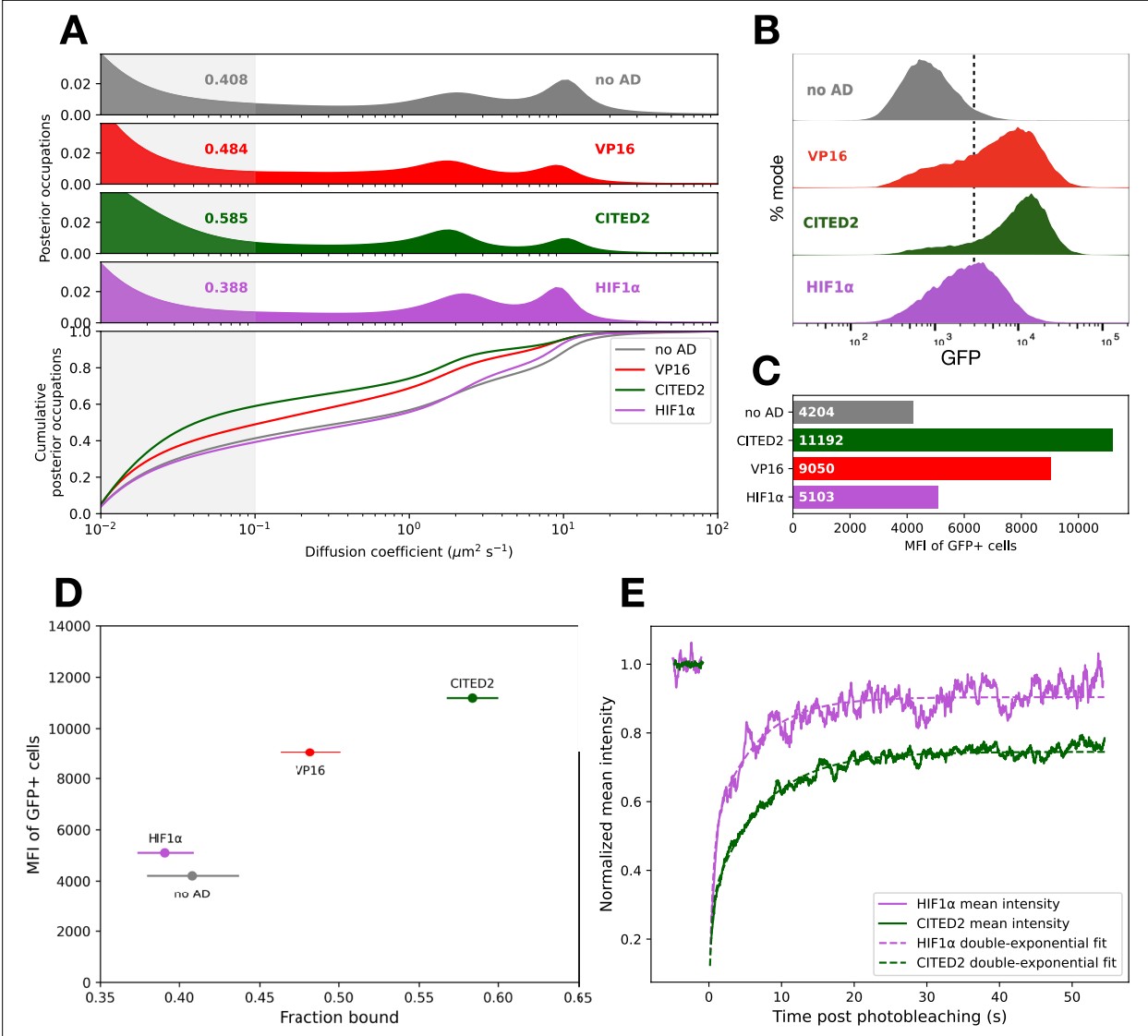

**Figure 2.** Short activation domains are sufficient to control the fraction of transcription factor (TF) molecules bound to chromatin. (**A**) Diffusion spectra (top) and cumulative distributions (bottom) of synthetic TFs bearing various activation domains. Numbers indicate the fraction of molecules with diffusion coefficients below 0.1 µm²/s (shaded region), which we infer to be chromatin-bound. (**B**) Flow cytometry measurements of cells bearing various activation domains after 24 hr of estradiol treatment. Distributions are shown in the top panel, where the dashed line is an arbitrary gate for GFP-positive cells. (**C**) Bar plot for mean fluorescence intensity (MFI), the geometric mean of GFP values for cells in the positive gate. (**D**) For each synthetic TF, the MFI of GFP+ cells is plotted against its bound fraction. Error bars indicate bootstrapping standard deviations of the bound fraction (Materials and methods). (**E**) HIF1α and CITED2 fluorescence recovery after photobleaching (FRAP) recoveries over time plotted as rolling means (solid lines). Curve fits to a double-exponential equation (Materials and methods) are shown in dashed lines.

The online version of this article includes the following figure supplement(s) for figure 2:

**Figure supplement 1.** Jump length histograms and Spot-On fits for two synthetic transcription factors (TFs).

**Figure supplement 2.** Analysis of a single-molecule tracking (SMT) dataset for sources of variance.

**Figure supplement 3.** Diffusion spectra of activation domain-synthetic transcription factor (TF) constructs in WT U2OS cells.

**Figure supplement 4.** Fluorescence recovery after photobleaching (FRAP) curves for histone H2B, HIF1α-synthetic transcription factor (TF), and CITED2-synthetic TF.

SMT at fast frame rates provides snapshots of the diffusive states of these factors but cannot accurately determine how long a given factor remains bound. To complement this experiment, we measured TF residence time on chromatin with FRAP (*Figure 2E*, *Figure 2—figure supplement 4*). We fit our FRAP recovery curves to a double-exponential equation (Materials and methods, *Supplementary*

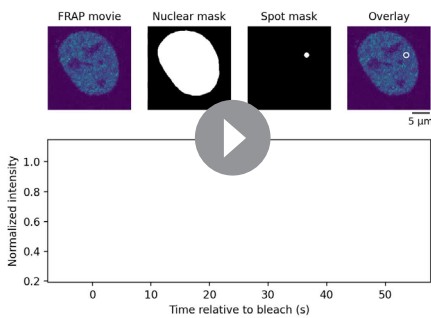

**Video 2.** Exemplary fluorescence recovery after photobleaching (FRAP) movie of a cell expressing CITED2-synthetic transcription factor (TF). FRAP movie, nuclear mask, photobleach spot mask, and an overlay of the movie and spot mask are shown above the normalized intensity of the FRAP recovery over time. Movies of many cells expressing CITED2-synthetic TF are aggregated and fit to generate a recovery curve, like that in *Figure 2E* (green).

https://elifesciences.org/articles/105776/figures#video2

Within each allelic series, we saw a positive correlation between activation domain strength and bound fraction. Stronger activation domain mutants had higher bound fractions. These mutations change activity by perturbing different physicochemical parameters. For CITED2 (*Figure 3A*), two alleles that decrease activity, either by mutating key hydrophobic residues (motif mutation) or replacing all six leucines with phenylalanines (L>F), decreased bound fraction. A CITED2 allele that neutralizes a single positive charge (K>A) increased activity and increased bound fraction. For VP16, two alleles that abrogate activation domain activity, F442A (*Cress and Triezenberg, 1991*) and 7As, both reduced the bound fraction (*Figure 3B*). In HIF1α, adding acidity by mutating three arginines to aspartates (R>D) or two glutamines and one asparagine to glutamate (QN>E) increased the frac-

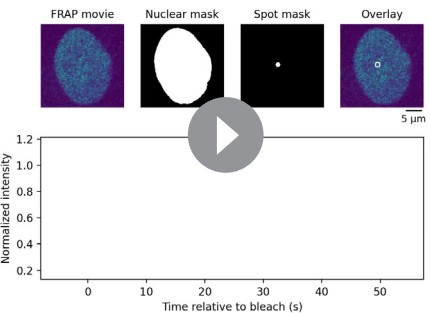

**Video 3.** Exemplary fluorescence recovery after photobleaching (FRAP) movie of a cell expressing HIF1α-synthetic transcription factor (TF). FRAP movie, nuclear mask, photobleach spot mask, and an overlay of the movie and spot mask are shown above the normalized intensity of the FRAP recovery over time. Movies of many cells expressing HIF1α-synthetic TF are aggregated and fit to generate a recovery curve, like that in *Figure 2E* (purple).

https://elifesciences.org/articles/105776/figures#video3

*file 1*, *Mueller et al., 2012*). We found that the SMT-bound fraction correlated with FRAP recovery times, with stronger activation domains conferring longer dwell times (*Videos 2 and 3*). This recovery time difference is consistent with another synthetic system where strong activation domains have longer dwell time on chromatin (*Trojanowski et al., 2022*). Together, these results indicate that stronger activation domains have a higher fraction of molecules bound to chromatin and that these molecules have longer residence times on chromatin.

## Mutations in activation domains modulate chromatin binding

Intrigued by the apparent correlation between activation domain strength and bound fraction, we investigated multiple allelic series of mutations that increased and decreased activity of our activation domains (*Staller et al., 2022b*). We selected mutants that we have previously shown to have large effects on reporter gene activity and that change one to seven residues in each activation domain.

tion bound (*Figure 3C*). In the SREBP activation domain, removing aromatic and/or leucine residues decreased activity and decreased the bound fraction (*Figure 3—figure supplement 1*). We employed a bootstrapping approach to estimate error in SMT diffusion spectra (Materials and methods, *Figure 3—figure supplement 2*) and noted that this trend held within each allelic series and across the full set of mutants (*Figure 3D*). In individual cells, the bound fraction was not correlated with TF abundance (*Figure 3—figure supplement 3*) or the charge of the activation domain (*Figure 3—figure supplement 4*). Within each allelic series, stronger activation domain mutants had higher bound fractions, and weaker mutants had lower bound fractions.

Furthermore, for HIF1α, the QN>E allele had slower FRAP recovery, indicating a longer residence time on chromatin (*Figure 3—figure supplement 5D*). These data support the idea

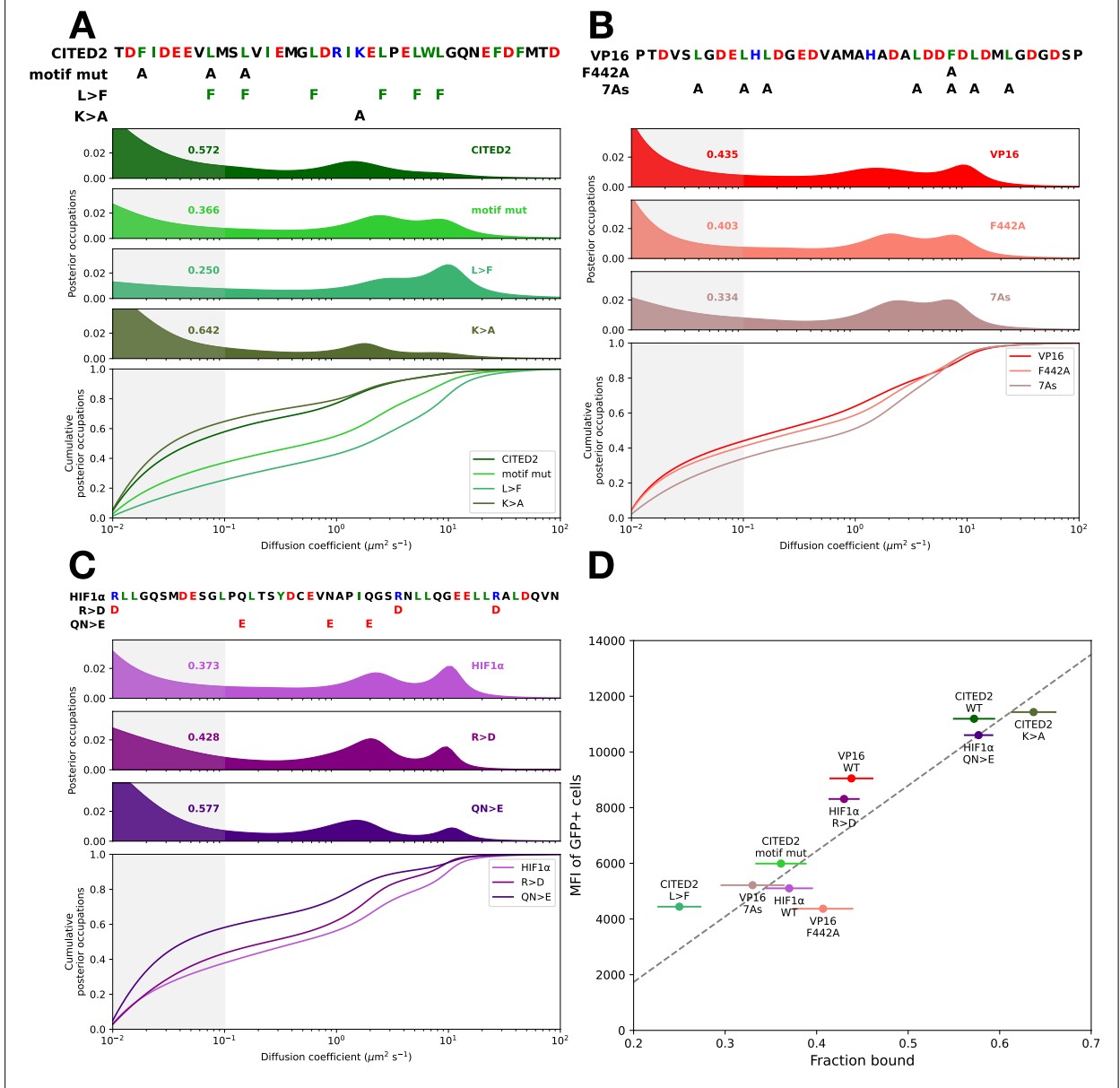

**Figure 3.** Mutations in individual activation domains modulate the fraction of molecules bound to chromatin. (**A**) The amino acid sequence of CITED2 and its mutant constructs is shown above their diffusion spectra and the cumulative distributions. Numbers indicate the fractions bound, the probability density in the shaded region of the diffusion spectra. (**B**) As in A for VP16. (**C**) As in A for HIF1α. (**D**) Summary of activation domain activity and chromatin-bound fractions. Mean fluorescence intensity of GFP+ cells plotted against bound fractions. Error bars indicate standard deviations of a cell-wise bootstrapping scheme (Materials and methods, *Figure 3—figure supplement 2*).

The online version of this article includes the following figure supplement(s) for figure 3:

**Figure supplement 1.** An additional allelic series with cholesterol transcription factor (TF) SREBP's activation domain.

**Figure supplement 2.** Example outputs from a cell-wise bootstrapping scheme to estimate error in single-molecule tracking (SMT) diffusion spectra.

**Figure supplement 3.** Cell-wise relationships between chromatin-bound fraction, transcription factor (TF) expression, and reporter output.

**Figure supplement 4.** Activation domain net charge and fractions bound for constructs in *Figure 3*.

**Figure supplement 5.** Mutations in activation domains change reporter activity and fluorescence recovery after photobleaching (FRAP) recovery.

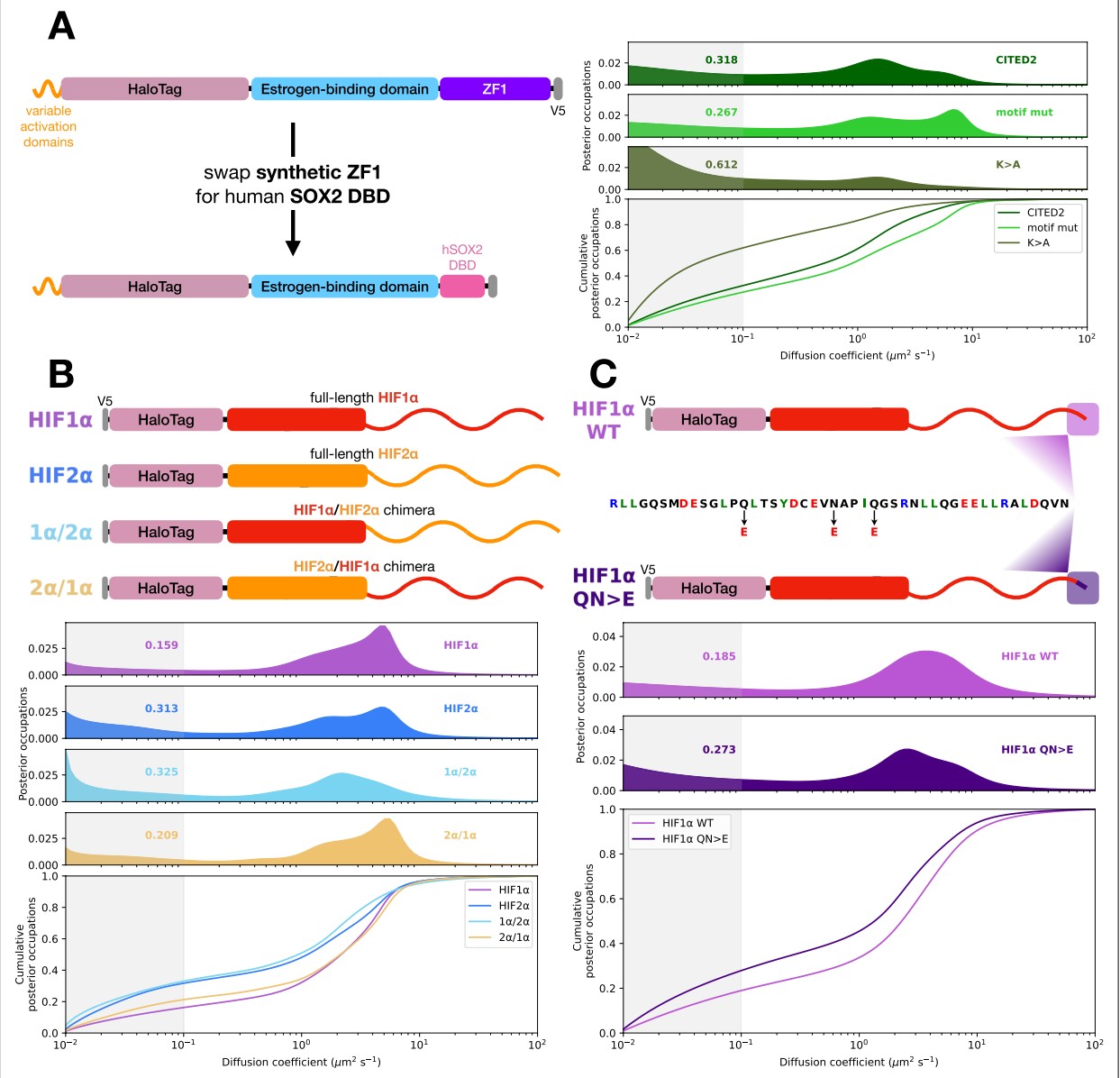

**Figure 4.** Activation domain alleles control the fraction of molecules bound to chromatin when fused to an alternative DNA-binding domain (DBD) or in a full-length transcription factor (TF). (**A**) Left: We replaced the synthetic zinc finger DBD with the human SOX2 DBD. Right: Diffusion spectra and cumulative distributions for three alleles of CITED2. Shaded region indicates density contributing to the bound fraction. (**B**) Top: Schematic of the HIF1α and HIF2α intrinsically disordered region (IDR) swap chimeras from *Chen et al., 2022*. Bottom: Diffusion spectra and cumulative distributions of the constructs shown. The HIF1α IDRs control the fraction of molecules bound to chromatin under hypoxic conditions. (**C**) Top: We introduced three superactivating point mutations (QN>E) into full-length HIF1α's second activation domain. Bottom: Diffusion spectra and cumulative distributions of the constructs shown. These three mutations in HIF1α increase the fraction of molecules bound to chromatin, recapitulating the differences seen in synthetic TFs.

The online version of this article includes the following figure supplement(s) for figure 4:

**Figure supplement 1.** AAVS1-integrated GFP reporter with synthetic ZF-binding sites is not activated by SOX2DBD-activation domain constructs.

that strong activation domains increase the fraction of TF molecules bound to chromatin and extend the duration of their binding.

## Activation domains control chromatin binding of a second DBD

We next tested whether the observed relationship between activation domain strength and bound fraction depends on the DBD of our synthetic TF. We replaced the synthetic zinc fingers with the SOX2

DBD, which belongs to a different structural family, the high-mobility group (HMG) DBDs (*Figure 4A*). These domains are minor-groove binders with many specific cognate sites throughout the genome. For this orthogonal DBD, our CITED2-K>A mutant had a dramatically higher fraction bound than WT, while the inactive motif mutant modestly decreased chromatin association (*Figure 4A*). The L>F mutation in this chimeric factor prevented its localization to the nucleus, precluding SMT. None of these constructs activated our GFP reporter (*Figure 4—figure supplement 1*). This last finding further reinforces that most of the observed chromatin binding in SMT is not at the reporter locus. With this second DBD, strong activation domains show more chromatin binding.

## Activation domains control chromatin binding of full-length TFs

We next investigated activation domain mutations in full-length TFs. We previously showed that the bound fraction of the HIF1α and HIF2α paralogs is controlled by long IDRs, not their DBDs (*Chen et al., 2022*). In chimeric TFs, the IDR of HIF2α yields more chromatin binding than that of HIF1α. Previous studies were conducted in VHL-deficient renal carcinoma cells, which resulted in the stabilization of HIF proteins in this cell line (*Brodaczewska et al., 2016*). We recapitulated these results in our U2OS cells using deferoxamine, a drug that prevents HIF degradation and mimics aspects of hypoxia (*Guo et al., 2006*). In this condition, we reproduced the effect of IDR swaps on the diffusion of HIF factors: HIF2α had a higher bound fraction than HIF1α, and the HIF2α IDR was sufficient to confer this higher bound fraction (*Figure 4B*). Conversely, the HIF1α IDR conferred a lower bound fraction when attached to the HIF2α DBD. For these paralogs, the IDR controls the bound fraction.

We tested if the mutations in the activation domain could control the fraction of full-length HIF1α bound to chromatin. Introducing the strongest of these mutations, QN>E, into AD2 (C-terminal activation domain) of full-length HIF1α increased the fraction of molecules bound to chromatin (*Figure 4C*). This result shows that the strength of minimal activation domains in endogenous factors can control the fraction of molecules bound to chromatin. Further, this experiment demonstrated that the relationship between activation domain strength and bound fraction holds for another structurally distinct DBD, the basic helix-loop-helix (bHLH) family.

Together, we found that strong activation domains control the bound fraction of one synthetic and two natural DBDs. These DBDs are taken from three structurally distinct classes: zinc finger, HMG, and bHLH. We have further shown this result holds for both synthetic TFs and full-length endogenous TFs. We speculate that acidic activation domains may contribute to the fraction of molecules bound to chromatin for other full-length TFs.

## Quantifying activation domain coactivator interactions in vivo

We tested the hypothesis that protein-protein interactions between activation domains and coactivators can explain the positive correlation between activation domain strength and the fraction of molecules bound to chromatin. If increasing activation domain binding to known coactivator partners can increase TF chromatin binding, it would explain our results without invoking any new molecular mechanisms. In vitro, strong activation domains bind coactivators more tightly than weak ones (*DelRosso et al., 2024*; *Henderson et al., 2018*; *Sanborn et al., 2021*). In the nucleus, coactivators are limiting for transcriptional activity (*Gillespie et al., 2020*) and have long residence times on chromatin (*Cho et al., 2018*; *Sanborn et al., 2021*). For p300, the majority of molecules are bound to chromatin at steady state (*Ferrie et al., 2024*).

To assess more directly whether chromatin binding of synthetic TFs correlates with coactivator binding, we used PAPA, in which excitation of a 'sender' fluorophore reactivates a nearby 'receiver' fluorophore from a dark state, thereby revealing proximity between two labeled proteins (*Figure 5A*). To this end, we fused SNAP-tag2 (*Kühn et al., 2025*) to synthetic TFs in a U2OS cell line with HaloTagged endogenous p300 (*Ferrie et al., 2024*). After labeling SNAP-tag2 with the receiver fluorophore JFX650 and HaloTag-p300 with the sender fluorophore JF549, we placed JFX650 in a dark state with 639 nm illumination and assayed proximity-dependent reactivation following excitation of JF549 with 561 nm light. As previously described, we used direct reactivation of JFX650 with 405 nm light as an internal normalization standard (*Dahal et al., 2025*; *Graham et al., 2025*; *Graham et al., 2022*) and calculated a normalized PAPA ratio relative to nuclear SNAP-tag2 alone, a noninteracting negative control (*Abidi et al., 2025*) (Materials and methods, *Figure 5—figure supplement 1*). A very marginal PAPA signal was observed between p300 and the synthetic TF without an activation domain, while a

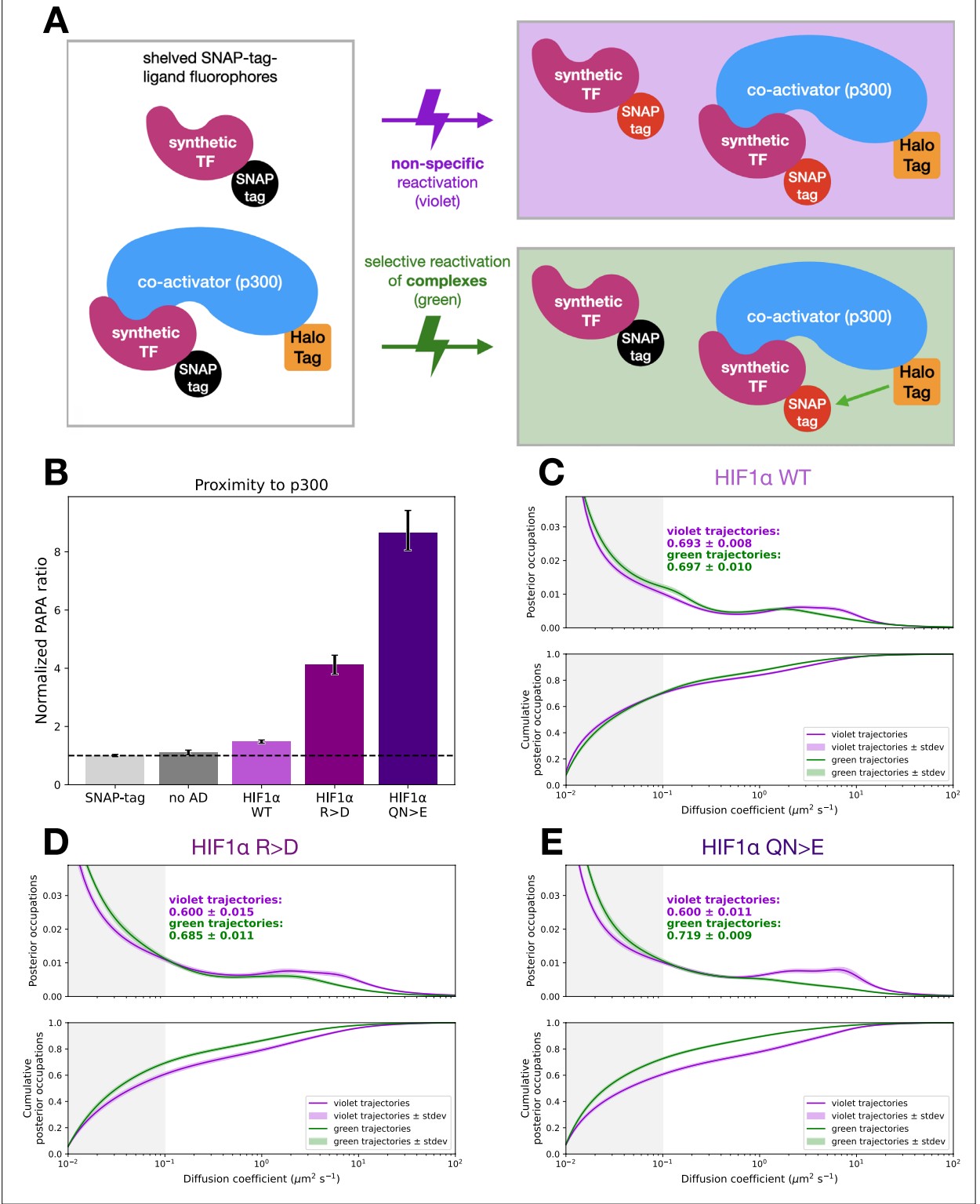

**Figure 5.** Proximity-assisted photoactivation (PAPA) of synthetic transcription factors (TFs) shows that stronger activation domains promote interaction with p300. (**A**) Schematic of our PAPA experiment using p300 endogenously tagged with HaloTag and an exogenous SNAP-tagged synthetic TF. First, red SNAP-tag2-liganded dyes are shelved (left, Materials and methods). A pulse of violet light (405 nm) indiscriminately reactivates these shelved dyes, while green light (561 nm) excites the sender, HaloTag-liganded dye, which reactivates nearby SNAP-tag dyes selectively. (**B**) Normalized PAPA ratio quantifying the proximity between p300-HaloTag and the constructs labeled. Error bars are 95% confidence intervals derived by bootstrapping (Materials and methods). (**C**) Diffusion spectra and cumulative distributions after segregation of green-reactivated (p300 proximal molecules) and violet-

*Figure 5 continued on next page*

*Figure 5 continued*

reactivated (direct reactivation) trajectories for cells expressing HIF1α WT. Density in the gray shaded region is quantified as the bound fraction. Light green and purple shading denotes the standard deviation of bootstrapping trials. (**D**) As in (**C**) for HIF1α RK >D. (**E**) As in (**C**) for HIF1α QN >E.

The online version of this article includes the following figure supplement(s) for figure 5:

**Figure supplement 1.** Proximity-assisted photoactivation (PAPA) analysis intermediates.

small but significant increase was observed upon fusion to the HIF1α activation domain (*Figure 5B*). A dramatic increase in PAPA signal was observed for the superactive mutants, consistent with stronger binding to p300. These data support the model that activation domains increase chromatin binding of TFs by engaging coactivators. These results also demonstrate that PAPA can detect interactions between a coactivator and minimal activation domains and that this measure is commensurate with activation domain strength.

In addition, we can examine the diffusion spectra of the p300 proximal molecules (green-reactivated) and the total population (violet-reactivated). For the two superactive mutants, this analysis shows that the p300 proximal population has a much higher bound fraction (*Figure 5D and E*). Increased binding between synthetic TFs and p300 measured by PAPA correlated with a higher chromatin-bound fraction in the SMT. Together, these results support the idea that activation domain coactivator binding can tether TFs to chromatin.

## Disrupting coactivator function perturbs TF-chromatin associations

We further investigated if activation domain binding to coactivators can tether TFs to chromatin by pharmacologically perturbing coactivator function. First, we used norstictic acid to selectively disrupt VP16 binding to the Med25 domain of Mediator (*Garlick et al., 2021*). Like most coactivators, Mediator has a longer dwell time on DNA than that of sequence-specific TFs (*Cho et al., 2018*; *Whitney and Lionnet, 2024*). If the synthetic TF with VP16 is tethered to chromatin via activation domain interactions with Med25, norstictic acid treatment would decrease the fraction of molecules bound to chromatin. In line with this prediction, norstictic acid caused a small but significant decrease in the bound fraction of synthetic TF with VP16 (*Figure 6A*). The empty TF experienced no significant change in binding (*Figure 6B*), and two alleles of VP16 also showed decreases in fraction bound in response to norstictic acid (*Figure 6—figure supplement 1*). This perturbation further supports the idea that synthetic TFs can be tethered to chromatin by activation domain interactions with coactivators.

For the second pharmacological perturbation, we selected the coactivator p300, which has a high bound fraction, a relatively long residence time (*Ferrie et al., 2024*), and is a key binding partner for the HIF1α, CITED2, and VP16 activation domains (*Dyson and Wright, 2016*). Inhibition of p300 with A485 inhibits catalytic activity, decreases global transcription (*Lasko et al., 2017*), and increases the fraction of p300 molecules bound to chromatin (*Ferrie et al., 2024*). Under A485, we predict that the same steady-state binding between an activation domain and p300 will result in more synthetic TF molecules binding to p300 on chromatin and give the appearance of more synthetic TF binding to chromatin. Consistent with this prediction, A485 treatment mildly but significantly increased the fraction of synthetic TF with VP16 bound to chromatin relative to control (*Figure 6C*). The empty synthetic TF experienced the opposite effect, with a decrease in binding to chromatin (*Figure 6D*). Together, the A485 and norstictic acid perturbations support the idea that TFs can be tethered to chromatin through protein-protein interactions with coactivators.

As a third perturbation of coactivator function, we inhibited the BAF (mammalian Swi/Snf) complex with BAFi (BRM014). BAF inhibition leads to rapid genome-wide chromatin compaction and decreased global transcription (*Schick et al., 2021*). In single-molecule footprinting experiments, inhibiting BAF decreased promoter occupancy of a Tet-VP48 synthetic TF (*Doughty et al., 2024*). For our synthetic TF, inhibition of BAF significantly decreased the fraction of molecules bound to chromatin relative to the vehicle control (*Figure 6E*). The empty synthetic TF did not experience a significant change in binding (*Figure 6F*). This result suggests that a substantial portion of our immobile TFs are binding chromatin at open or actively transcribing loci.

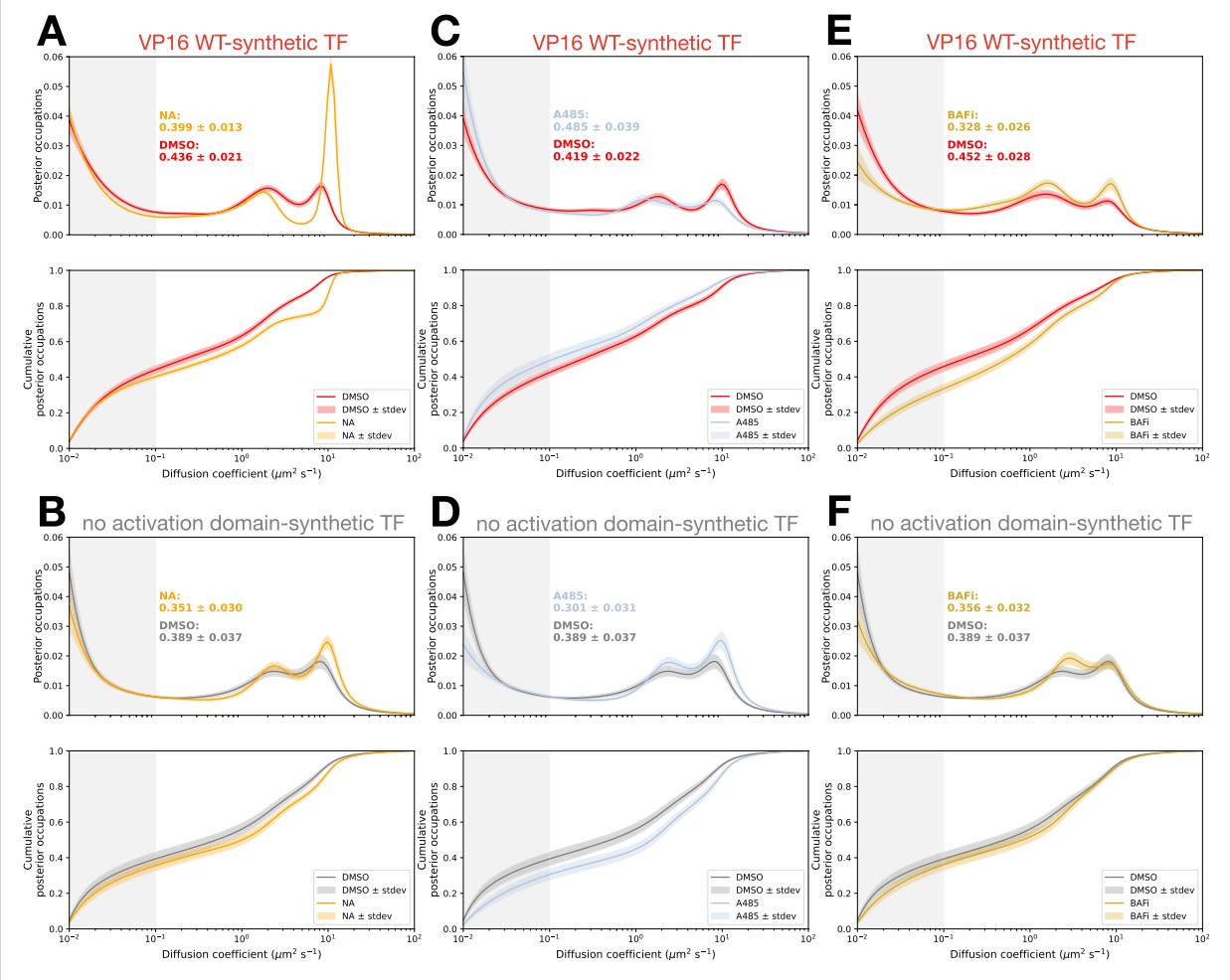

**Figure 6.** Perturbing coactivator function alters transcription factor (TF) bound fraction. (**A**) VP16-synthetic TF diffusion spectra and cumulative distributions for cells treated for 2 hr with norstictic acid or vehicle (DMSO). Density in light gray is quantified as the bound fraction. Shading around the spectra and cumulative distributions indicates standard deviation of bootstrapping trials. (**B**) Diffusion spectra and cumulative distributions of cells expressing the synthetic TF with no activation domain treated for 2 hr with norstictic acid or vehicle (DMSO). Density in light gray is quantified as the bound fraction. Shading around the spectra and cumulative distributions indicates standard deviation of bootstrapping trials. (**C**) As in (**A**) for A485, a p300 inhibitor. (**D**) As in (**B**) for A485, a p300 inhibitor. The DMSO curve is repeated from (**B**). (**E**) As in (**A**) for BRM-014, a BAF inhibitor (BAFi). (**F**) As in (**B**) for BRM-014, a BAF inhibitor (BAFi). The DMSO curve is repeated from (**B**).

The online version of this article includes the following figure supplement(s) for figure 6:

**Figure supplement 1.** Drug perturbations on the synthetic transcription factor (TF) bearing mutants of the VP16 activation domain.

## Synthetic TFs bind active loci genome-wide

The increases in chromatin binding measured by SMT cannot distinguish between stronger binding to the same genomic loci and binding to new genomic loci. To differentiate between these two possibilities, we undertook CUT&RUN experiments (*Skene and Henikoff, 2017*) on five of our synthetic TFs (*Figure 7A*). For full-length HIF1α and HIF1β TFs, CUT&RUN experiments indicated most genomic loci are bound by both TFs, implying that an increased bound fraction in SMT experiments is primarily due to higher occupancy at the same loci (*Chen et al., 2022*). This result sets the expectation that our activation domain mutations that increase chromatin-bound fraction in SMT are unlikely to bind new genomic loci and instead will have higher occupancy at the same loci across the genome.

All the TFs, including the empty TF, were bound to the cognate binding sites of the reporter gene at the AAVS1 locus. Normalizing between samples revealed that the HIF1α QN>E superactive mutant had more occupancy than the WT HIF1α TF (*Figure 7C*). Genome-wide, the binding profiles of all synthetic TFs mostly overlapped pre-existing ATAC-seq peaks (*Figure 7D*, *Figure 7—figure*

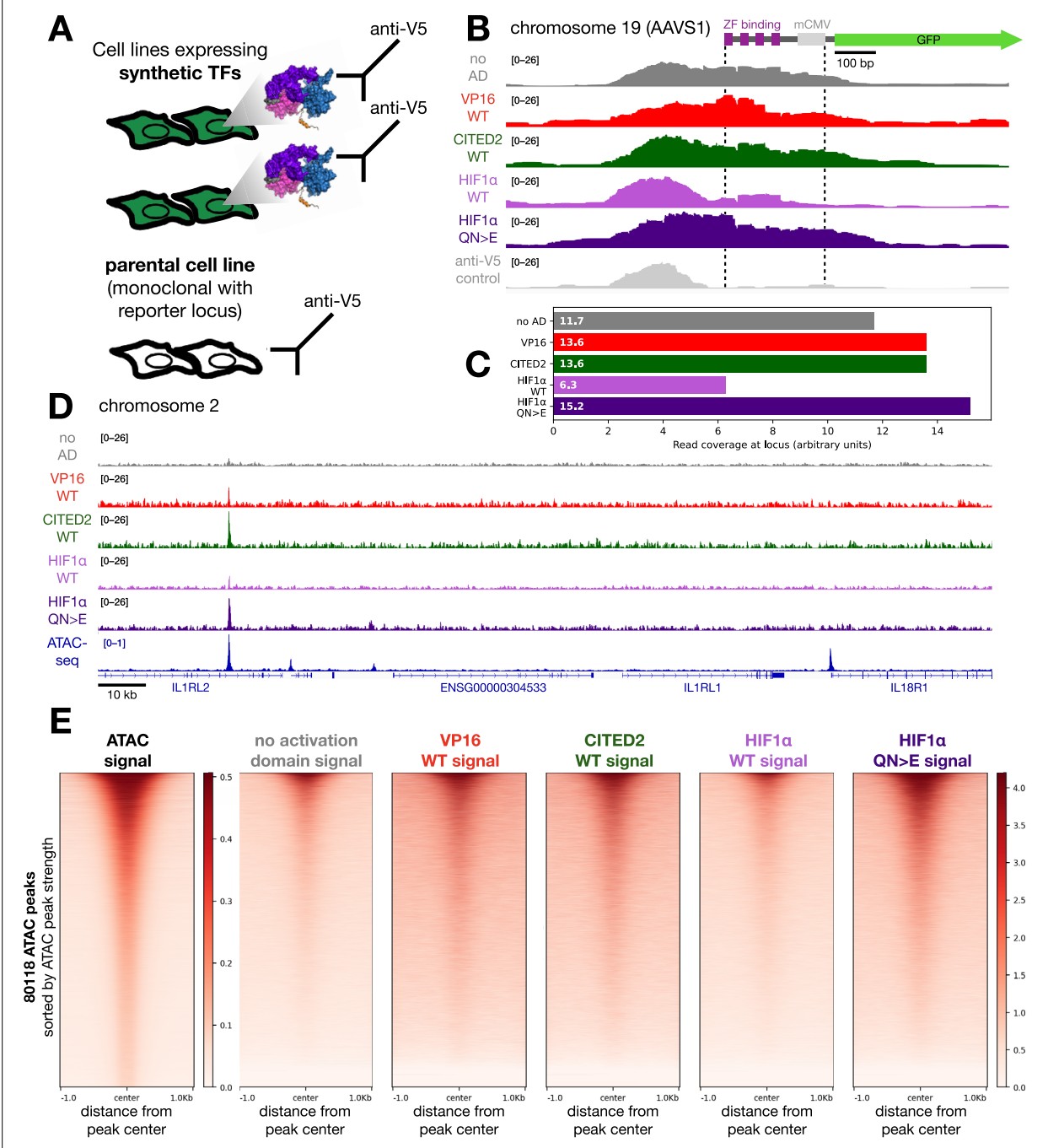

**Figure 7.** CUT&RUN on synthetic transcription factors (TFs) with different activation domains reveals most binding occurs at open chromatin. (**A**) We performed CUT&RUN on the parental cell line bearing the GFP reporter locus and five cell lines, each expressing synthetic TFs with a variable activation domain. Cells for all conditions were probed with an anti-V5 antibody, which should bind specifically only to the V5 epitope of the synthetic TFs. (**B**) Genomic binding signal at the engineered AAVS1 locus, shown to scale. All the synthetic TFs, including the empty TF, bound the cognate binding sites in the reporter construct. (**C**) Mean binding signal of the various synthetic TFs between the dashed lines of (**B**), which include the ZF1 binding sites and minimal promoter. (**D**) Binding of synthetic TFs and ATAC-seq signal at an exemplary locus. TF binding signal is strongest at ATAC-seq peaks, with stronger activation domains conferring more genomic binding in some cases. Not all ATAC-seq peaks are bound by the synthetic TFs. (**E**) ATAC-seq (left) and synthetic TF factor binding (right plots) signals plotted across 80118 ATAC-seq peaks, sorted by decreasing ATAC-seq peak strength. The five synthetic TFs share a color scale; the ATAC-seq signal uses a separate scale. Binding patterns for the synthetic TFs generally follow the strength of the ATAC-seq signal.

The online version of this article includes the following figure supplement(s) for figure 7:

**Figure supplement 1.** Synthetic transcription factors (TFs) bearing various activation domains generally bind to similar, open regions of the genome.

*supplement 1A*). The HIF1α superactive allele bound more loci than the WT allele, but the patterns were highly overlapping (*Figure 7—figure supplement 1B*). Together, these results indicate that most synthetic TF binding events occur at accessible loci. We conclude that the increased bound fraction we observe in the SMT primarily results from more binding to the same active loci.

## Discussion

How a TF binds the genome is controlled by both DBDs and IDRs, but it remains difficult to predict the relative contributions of these two regions. We demonstrated that very short activation domains can control the fraction of TF molecules bound to chromatin, sometimes accounting for more binding than the DBD. Mutations that increase activation domain strength in reporter assays increase the fraction of molecules bound to chromatin. Conversely, mutations that decrease activation domain strength decrease the fraction of molecules bound to chromatin. This trend holds for allelic series of four acidic activation domains, for three structurally diverse DBDs, and for synthetic and full-length TFs.

SMT has revealed many new dimensions of TF nuclear dynamics and has been instrumental for our discovery. The technique has shown that TF-chromatin interactions are generally short-lived (*Brouwer and Lenstra, 2019*; *Lionnet and Wu, 2021*; *Liu et al., 2014*), that coactivator chromatin binding is long-lived (*Ferrie et al., 2024*), and that IDRs can make larger contributions to chromatin binding than DBDs (*Chen et al., 2022*). Here, we demonstrate that minimal activation domains (39–60AA) can control the fraction of molecules bound to chromatin. Our results are consistent with an SMT study of an activation domain deletion in p53 showing reduced association with dense chromatin (*Mazzocca et al., 2023*).

There are three hypotheses for how increasing activation domain strength causes a higher fraction of TF molecules bound to chromatin: (1) activation domains could directly bind DNA, (2) activation domains could modulate DBD interactions with DNA, or (3) activation domains could bind to coactivators that are bound to chromatin, tethering the TF to chromatin.

Our short acidic activation domains are unlikely to bind DNA directly due to electrostatic repulsion: both the activation domains and DNA are negatively charged and repel each other. While protein net charge can play a large role in nuclear diffusion (*Xiang et al., 2020*), we found no correlation between net charge and fractions bound in factors we tested (*Figure 3—figure supplement 4*).

It is also possible that our activation domains are interacting with the DBD to modulate DNA binding, an emerging theme in TF biology (*Baughman et al., 2022*; *Bentley et al., 2023*; *Bjarnason et al., 2024*; *He et al., 2019*; *Krois et al., 2018*; *Sun et al., 2021*). For our results, intramolecular interactions are not the main drivers of chromatin binding because the phenotype is consistent across three structurally distinct DBDs.

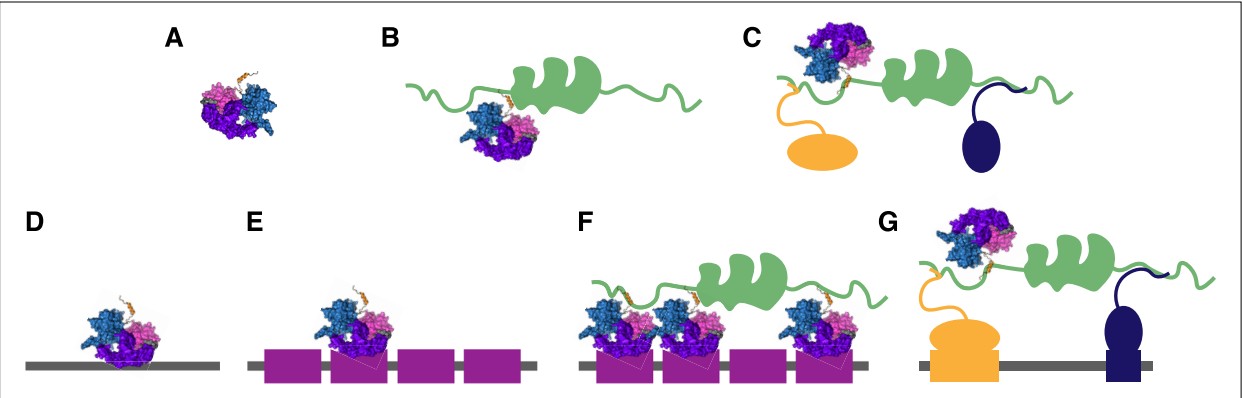

**Figure 8.** Model for how activation domains contribute to chromatin binding by interactions with coactivators. (**A**) A solitary transcription factor (TF) diffusing in the nucleoplasm, a lone wolf. (**B**) A TF bound to a coactivator with its activation domain diffusing through the nucleoplasm. (**C**) Three TFs bound simultaneously to a coactivator diffusing in the nucleoplasm as a complex or wolfpack. (**D**) A TF nonspecifically bound to random DNA. (**E**) A TF specifically bound with its DBD to a cognate DNA motif at the reporter locus. (**F**) TFs specifically bound to motifs at the reporter locus and simultaneously bound to a coactivator. (**G**) Proposed model for how the activation domain tethers the synthetic TF to chromatin through the coactivator. This situation occurs at active loci where the coactivator has been recruited by the orange and blue TFs.

Our PAPA experiments support the hypothesis that protein-protein interactions between activation domains and coactivators explain the positive correlation between activation domain strength and the fraction of TF molecules bound to chromatin. The overall chromatin-bound fraction of SNAP-tagged synthetic TFs was higher than that of the Halo-tagged versions, perhaps reflecting differences in physical properties of the two tags, such as volume and charge (*Figure 3C* vs *Figure 5C–E*, violet curves). TFs reactivated by PAPA from p300 had a higher chromatin-bound fraction than those directly reactivated by violet light (DR), indicating the expected correlation between p300 binding and chromatin binding (*Figure 5*). Compared to WT, the two superactive HIF1α alleles showed stronger binding to p300 by PAPA and a more pronounced shift toward chromatin-bound molecules when comparing PAPA to DR trajectories. Curiously, while the bound fraction of PAPA trajectories increased with activation domain strength, DR trajectories, if anything, showed the opposite trend. One possible explanation is that a large fraction of TF molecules bind p300, and each round of PAPA followed by re-bleaching depletes these complexes from subsequent rounds of direct reactivation—with more interaction resulting in greater depletion. This work marks the first time that PAPA has been applied to measure TF-coactivator interactions in vivo, which we anticipate will be a powerful new experimental approach to study transcription. Together with the PAPA experiments, the pharmacological perturbations support our hypothesis that activation domain-coactivator interactions strongly influence TF binding to chromatin (*Figure 8*). We cannot distinguish whether the coactivator directly binds DNA, binds a nucleosome, or binds another TF bound to DNA, but we suspect all three binding modes are involved.

In the CUT&RUN data, stronger activation domains cause higher occupancy at the reporter locus, but the vast majority of other genomic binding occurs at other active loci. Consequently, we believe our activation domain-mediated chromatin binding is distinct from the ability of long IDRs to direct TFs to specific genomic loci (*Brodsky et al., 2020*; *Gera et al., 2022*; *Goldman et al., 2023*; *Hurieva et al., 2024*; *Mindel et al., 2024a*). Instead, we believe the SMT is observing the same underlying biological process that causes activation domains to enhance promoter occupancy in single-molecule footprinting experiments (*Doughty et al., 2024*) and long TF residence times on promoters (*Schaepe et al., 2025*). It is remarkable that short activation domains can have such a large effect on binding.

Genome binding is an emergent property of full-length TFs, with contributions from the DBD, activation domains, and their interactions. In textbook models, the DBD binds DNA first, and then the activation domain recruits the coactivators. We speculate that recruitment can occur in both directions: sometimes the first event will be the activation domain binding to a coactivator that is already engaged at an active locus, and then the DBD will bind the local DNA (*Figure 8*). Different activation domains bind distinct coactivators (*Alerasool et al., 2022*), but only some of these interactions, which can have affinities of 10 nanomolar (*Berlow et al., 2017*; *De Guzman et al., 2004*), will be strong enough to tether TFs to chromatin.

Our results also challenge the traditional interpretation of activation domain strength. Previously, we had assumed that all our synthetic TFs had similar DNA occupancy because they have the same DBD. Our results here show that strong activation domains increase occupancy at a reporter locus. As a result, the higher reporter activity is a convolution of more binding and more coactivator recruitment (*Doughty et al., 2024*). This finding motivates the need for new assays for activation domain function that can separate binding from reporter activity.

In this work, we focused on the fraction of molecules bound to chromatin. We also see some changes in the distribution of inferred diffusion constants of mobile molecules, some of which are bimodal. After norstictic acid treatment, the slow-moving population shrank and the fast-moving population grew. The slow-moving population is also prominent in the PAPA data. We speculate that the slow-moving population might be TFs bound to coactivators exploring the nucleus in clusters or 'wolf packs' (*Figure 8C*), as we previously hypothesized (*Staller, 2022a*).

## Materials and methods
### U2OS cell culture
U2OS cells (osteosarcoma from a 15-year-old human female) were originally obtained from the UC Berkeley Cell Culture Facility. Cells were maintained at 37°C in a humidified atmosphere with 5% $CO_2$ and cultured in reconstituted DMEM containing 4.5 g/L glucose (Thermo 12800082) supplemented

with 9.1% fetal bovine serum (FBS, HyClone), L-glutamine (Sigma), GlutaMax (Thermo), and 100 U/mL penicillin-streptomycin (Thermo). Cells were generally passaged every 3 days by splitting 1:8, using 0.25% trypsin to facilitate detachment.

Cell lines used in this study tested negative for mycoplasma about monthly by PCR (https://bite-sizebio.com/23682/homemade-pcr-test-for-mycoplasma-contamination/). Cells were tested to be U2OS cells with STR profiling at the UC Berkeley Cell Culture Facility.

For imaging experiments, cells were plated the evening prior to imaging to achieve 60–80% confluency on the day of imaging. Cells were grown on glass-bottom dishes (MatTek P35G-1.5-14-C) in phenol-red-free DMEM containing 4.5 g/L glucose and supplemented with 9% FBS and 100 U/mL penicillin-streptomycin ('imaging medium').

## U2OS cell line generation

To generate the reporter cell line, U2OS cells were transfected in six-well dishes using Lipofectamine 3000, with 3 µg of DNA and 6 µL of Lipofectamine 3000 per well. Each well received 1 µg of a combined Cas9 and sgRNA plasmid targeting the AAVS1 locus (Addgene plasmid #227272) and 2 µg of repair template containing the reporter locus, similar to that of the previously described construct (*Staller et al., 2022b*).

The reporter construct is based on a published reporter (Addgene #176294), but we replaced puromycin resistance with blasticidin resistance, added VKYY to the start of the GFP to boost translation (*Verma et al., 2019*), inserted a small t intron, and inserted a WPRE before the SV40 poly(A) signal.

Two days after transfection, cells were passaged onto a 10 cm dish and selected with blasticidin (Fisher A1113903) at 5 µg/mL. When this dish was confluent, single clones were isolated by limiting dilution into 96-well plates and sequence-validated to have at least one copy of the reporter locus by PCR and Sanger sequencing. Clones were also tested by transient transfection of a VP16 synthetic TF by measuring fold-induction in response to estradiol induction. We picked the most robustly induced clone for this study. Cells were maintained in media containing 5 µg/mL blasticidin.

All synthetic zinc-finger- and SOX2DBD-bearing constructs were cloned using Gibson Assembly and verified via whole-plasmid sequencing (Plasmidsaurus). Starting with a published synthetic TF (Addgene #176293), we moved the TF into PiggyBac backbone with a L30 promoter, replaced mScarlet with HaloTag, added a V5 tag, and replaced the neomycin resistance with puromycin resistance. These constructs were stably integrated into the reporter clone described above. For integrations, we transfected 2 µg of a synthetic construct with 1 µg of a vector expressing Super PiggyBac Transposase (*Yusa et al., 2011*) in this reporter clone. Two days after transfection, we passaged cells onto a 10 cm dish and selected cells with puromycin (Invivogen) at 1 µg/mL. We continued passaging these cells for at least 2 weeks before proceeding with experiments. Cells were maintained in media containing 5 µL/mL blasticidin and 1 µg/mL puromycin.

## Selection of activation domain mutations

We selected mutations that strongly changed activation domain activity in our published assay (*Staller et al., 2022b*). The QN>E mutation prevents hydroxylation, a modification which normally destabilizes the protein and interferes with p300 binding (*Hewitson et al., 2002*; *Lando et al., 2002a*; *Lando et al., 2002b*).

## Nucleofection and drug treatments

For *Figure 4B and C*, U2OS cells were electroporated using a homemade buffer (*Danner, 2020*) and the X-001 protocol of the Lonza Nucleofector II. This method mitigates autofluorescence artifacts from lipid-based transfection reagents. One million U2OS cells were electroporated with previously described constructs (*Chen et al., 2022*) and treated with deferoxamine mesylate salt (Sigma D9533) at 100 µM overnight.

For *Figure 5A, D*, *Figure 5—figure supplement 1*, uninduced cells were treated for 2 hr with 10 µM norstictic acid (Santa Cruz Biotechnology) or the equivalent DMSO. These cells were stained with HaloTag ligand (as in 'Sample preparation for imaging experiments') and induced with 1 µM β-Estradiol (Sigma E2758; 'estradiol') immediately before imaging.

For *Figure 5B and E*, uninduced cells were treated for 2 hr with 10 μM A485 (Fisher) or the equivalent DMSO. These cells were stained with HaloTag ligand (as in 'Sample preparation for imaging experiments') and induced with 1 μM estradiol immediately before imaging.

For *Figure 5C and F*, uninduced cells were treated for 2 hr with 10 μM BRM/BRG1 ATP Inhibitor-1 compound 14 (MedChemExpress; 'BAFi') or the equivalent DMSO. These cells were stained with HaloTag ligand (as in 'Sample preparation for imaging experiments') and induced with 1 μM estradiol immediately before imaging.

Except for treatments noted above in this section, cells in all other conditions were treated with 1 μM estradiol for 12–16 hr the night before they were imaged for SMT.

For all flow cytometry experiments, cells were treated with 1 μM estradiol for 24 hr.

## Incucyte live-cell imaging

To assay reporter induction at fine time intervals, we used the Incucyte S3 system to acquire images every 2 hr post estradiol induction. The day before the experiment, cells were plated at around 25% confluence. The day of the experiment, cells were induced with 1 μM estradiol, and the first image was taken about 5 min later (as the 0 hr time point).

Images in phase and the green channel (to quantify GFP) were acquired with the 10× objective. The Incucyte S3 software was used to segment cells and quantify the amount of GFP in each cell. These amounts were used to draw an arbitrary gate separating GFP-positive and -negative cells. The percentage of GFP-positive cells was computed and plotted as a function of time using Python.

## Flow cytometry

The day of the assay, cells were stained for 10 min with 35 nM of JFX650-HaloTag ligand and 10 μM Hoechst 33342 (Thermo), rinsed once with imaging medium, and incubated in imaging medium for 25 min. Cells were lifted using 0.25% trypsin, quenched with imaging medium, and filtered through a cell strainer to remove clumps.

Flow cytometry was conducted on a BD Bioscience LSR Fortessa in the UC Berkeley Flow Cytometry Core. Conditions presented together as panels were collected on the same day with the same voltages. These voltages were not always kept consistent from day to day, as daily CST calibration often changed cytometer baselines. 30,000 measurements were made for each condition.

Data were analyzed using FlowJo, with consistent gating applied across conditions that are presented together in a panel. The reporter fluorescence distributions were frequently bimodal, so we quantified reporter activity in two ways. First, we used the fraction of cells that were GFP positive, but this metric had a limited dynamic range. Second, we used the mean fluorescence intensity of GFP-positive (GFP+) cells, which had a larger dynamic range.

## SMT movie acquisition

The day of imaging, cells were stained for 5 min with JFX650-HaloTag ligand (generously gifted by the Lavis lab), added to a final concentration of 5 nM and 2 μM Hoechst 33342 (Thermo), rinsed once with imaging medium, and incubated in imaging medium for 10 min. The dishes were rinsed one more time, replenished with imaging medium, and mounted on the microscope with Nikon immersion oil Type F.

The microscope used for SMT is as previously described (*Hansen et al., 2018*). In brief, this imaging setup consisted of a Nikon TI microscope fitted with a 100×/NA 1.49 oil-immersion TIRF objective, a motorized mirror, a Perfect Focus system, an EM-CCD camera, and an incubation chamber maintained at 37°C with a humidified 5% $CO_2$ atmosphere. The laser launch includes lasers of the following wavelengths: 405 nm (140 mW, OBIS, Coherent), 488 nm, 561 nm, and 639 nm (all 1 W, Genesis, Coherent). Laser intensities were regulated using an acousto-optic tunable filter (AA Opto-Electronic, AOTFnC-VIS-TN) and triggered via the camera's TTL exposure output signal. Lasers were delivered to the microscope through an optical fiber, reflected by a quad-band dichroic mirror (405 nm/488 nm/561 nm/633 nm, Semrock), and focused at the back focal plane of the objective. The incident laser angle was adjusted to achieve HILO illumination (*Tokunaga et al., 2008*). Emission light was filtered using the appropriate Semrock single band-pass filters.

For the diffusion spectra shown in each panel, all conditions were imaged on the same day to minimize day-to-day variability. A second replicate was imaged on a different day, and the data from both days were pooled to generate the reported spectra.

All experiments except those in *Figure 4B, C*, *Figure 2—figure supplement 3* used an automated imaging scheme derived from one previously described in *Walther et al., 2024*. Briefly, custom NIS Elements Macro Language and Python scripts instructed the microscope to raster over a coverslip, taking 81.92 μm × 81.92 μm images at each grid point. Cell presence in a field was determined using StarDist segmentation (*Schmidt et al., 2018*) in the Hoescht channel; images in additional channels of interest (GFP: 488 nm, JFX650: 639 nm) were also acquired. If cells were identified, one was randomly selected for imaging at 7.48 ms using the following protocol: 1200 frames with 2 ms stroboscopic illumination of the 639 nm laser (1 W, 100%), followed by 1200 frames with 2 ms stroboscopic illumination of the 639 nm laser (1 W, 100%) combined with 30% 405 nm laser power during camera transition intervals, and finally 1200 frames with 2 ms stroboscopic illumination of the 639 nm laser (1 W, 100%) combined with 60% 405 nm laser power during camera transition intervals. The 405 nm laser pulses during transition times were used to reactivate dark JFX650 fluorophores (*Endesfelder and Heilemann, 2015*) while minimizing background contribution by the laser pulse itself.

## PAPA SMT movie acquisition

The day of imaging, cells were stained for 10 min with 50 nM JFX549-HaloTag ligand and either 5 nM JFX650-SNAP-tag ligand (for SNAP-tag2 control conditions) or 50 nM JFX650-SNAP-tag ligand (for all other conditions; all dyes were generous gifts from the Lavis lab). Dishes were rinsed once with imaging medium and incubated in imaging medium for 15 min. The dishes were rinsed once more, replenished with imaging medium, and mounted on the microscope (same as described in 'SMT movie acquisition') with Nikon immersion oil Type F.

All PAPA experiments used the microscope setup and automation scheme previously described in 'SMT movie acquisition' with the following differences.

We used the 561 nm laser to measure p300-HaloTag nuclear intensities, and only cells with mean intensity values between 2000 and 20,000 were considered for imaging. StarDist-segmented objects with an area less than 3500 pixels (89.6 μm$^2$) or more than 7000 pixels (179.2 μm$^2$) were rejected. We also acquired images using the 639 nm laser for the SNAP-tagged proteins' expression levels and only considered nuclei with mean intensities above 5000. If more than one suitable cell was found for imaging, we selected the largest cell to image to recover the most localizations possible.

After a cell was selected by this automated scheme, the imaging macro shrank the region of interest to encapsulate the nucleus and applied the 639 nm laser at 100% intensity for 5 s to 'shelve' fluorescent SNAP-tagged molecules. Then, we used the following illumination sequence with eight phases (all 7.48 ms frames):

1. 200 frames of 1 W 639 nm laser at 100% intensity: these frames are not saved and serve to shelve SNAP-tagged molecules before the next reactivation pulse.
2. 30 frames of 1 ms stroboscopic 1 W 639 nm laser at 100% intensity: these frames are saved and are a readout for 'spontaneous reactivation', reactivation of molecules that is independent of a green or violet pulse.
3. 2 frames of 405 nm laser at 100% intensity: this pulse of violet light reactivates 'shelved' SNAP-tagged molecules regardless of whether or not they are complexed. These frames are not saved.
4. 30 frames of 1 ms stroboscopic 1 W 639 nm laser at 100% intensity: these frames are saved and read out the violet-reactivated molecules from the previous pulse.
5. 200 frames of 1 W 639 nm laser at 100% intensity: these frames are not saved and serve to shelve SNAP-tagged molecules before the next reactivation pulse.
6. 30 frames of 1 ms stroboscopic 1 W 639 nm laser at 100% intensity: these frames are saved and are a readout for 'spontaneous reactivation', reactivation of molecules that is independent of a green or violet pulse.
7. 50 frames of 100 mW 561 nm laser at 100% intensity: this pulse of green light reactivates SNAP-tagged molecules dependent on their proximity to p300-HaloTag. These frames are not saved.
8. 30 frames of 1 ms stroboscopic 1 W 639 nm laser at 100% intensity: these frames are saved and read out the green-reactivated molecules from the previous pulse.

This sequence is repeated five times before the cell-finding phase of the macro starts again.

## SMT and PAPA-SMT analysis

Spots were detected, subpixel-localized, and linked into trajectories using algorithms within the open-source Python package quot (https://github.com/alecheckert/quot, copy archived at *Heckert and Fan, 2025c*) using the following parameters:

'filter': {'start': 0, 'method': 'identity'}, 'detect': {'method': 'llr', 'k': 1.5, 'w': 11, 't': 18.0}, 'localize': {'method': 'ls_int_gaussian', 'window_size': 11, 'sigma': 1.5, 'ridge': 0.0001, 'max_iter': 10, 'damp': 0.3, 'camera_gain': 109.0, 'camera_bg': 470.0}, 'track': {'method': 'conservative', 'pixel_size_um': 0.16, 'search_radius': 1.0, 'max_blinks': 0, 'min_I0': 0.0}.

We segmented cell nuclei images with either StarDist (*Schmidt et al., 2018*) or cellpose (*Stringer et al., 2021*) as a starting point but found that manual QC and refining were necessary for accurate masking. This was done with a custom Python GUI (https://github.com/vinsfan368/qtpicker, copy archived at *Fan, 2025a*).

Only trajectories whose points lie entirely within one, and only one, curated mask were analyzed. For nuclei with more than six localizations per frame in at least one frame, we truncated frames from the beginning of the movie until all frames had fewer than seven localizations per frame. We then excluded cells with fewer than 100 displacements to ensure greater confidence in cell-wise posterior estimates. These three filters were post hoc choices and not pre-established exclusion criteria.

The resulting trajectories were analyzed using the open-source and previously published Python package saspt (https://github.com/alecheckert/saspt, copy archived at *Heckert et al., 2025a*; *Heckert et al., 2022*). We used the following inference parameters: focal_depth = 0.7, sample_size = 1,000,000, likelihood_type='rbme', splitsize = 3, start_frame = 0 (with the sample_size parameter being arbitrarily large to analyze all trajectories). For *Figure 4B and C*, we used the above parameters but replaced splitsize = 3 with splitsize = 8 to maintain consistency with our previously published results (*Chen et al., 2022*).

In figures where errors are reported, we followed a similar scheme previously described in *Ferrie et al., 2024*: for a dataset comprising n cells, we subsampled n random cells with replacement 96 times. For each of these trials, we pooled all trajectories from the n random cells, ran inference using the parameters described above, and noted the fractions bound. The bootstrapping means, standard deviations, and 95% confidence intervals are reported and/or plotted where indicated.

## Quantifying sources of variance in SMT data

To quantitatively describe sources of variance (*Figure 2—figure supplement 2*) within our data in a non-model-dependent and computationally inexpensive way, we were inspired by *Driouchi et al., 2025*. Briefly, we collate trajectories, noting the day on which they were imaged and the cell from which they were collected. We then decompose this data into individual displacements and subsample arbitrary numbers (n) of these jumps 1000 times, noting the mean value of these 1000 trials. By varying both n and the sampling scheme, we can assess sources of variance. We sampled according to three schemes: (1, gray) n jumps from the entire pool of trajectories, which establishes the baseline and should follow the law of large numbers as n becomes larger; (2, blue) by picking one cell from the dataset and sampling n jumps from only that cell; and (3, orange) by picking an imaging day from the dataset and sampling n jumps imaged only on that day. When curves 2 and 3 diverge from 1, all variance with respect to n has been captured. We note that variability between fields of view could contribute to cell-to-cell variability, but given that most of our fields of view contained only one cell, we did not quantify this source of variability. We found in *Figure 2—figure supplement 2* that cell-to-cell variability is about 100× higher than day-to-day variability.

## Normalized PAPA ratio and confidence intervals

To calculate the normalized PAPA ratio (*Figure 5B*), we tallied green- and violet-reactivated molecules in PAPA movies and additionally controlled for spontaneous reactivation of molecules: green- and violet-independent reactivations that likely depend on the expression level of the SNAP-tagged protein in a given nucleus. To do this, we subtracted the number of spontaneously reactivated molecules from the number of molecules reactivated by a preceding green or violet pulse; e.g., molecules localized in phase 2 of the illumination sequence in 'PAPA SMT movie acquisition' were subtracted from those in phase 4, and molecules localized in phase 6 of this sequence were subtracted from those in phase 8. This yielded a corrected green-to-violet (G/V) ratio for each cell measured.

Next, we divided these cell-wise corrected G/V ratios for each condition by the aggregate corrected G/V ratio of the SNAP-tag2 control imaged on the same day (*Figure 5—figure supplement 1D–F*). For example, if a cell expressing HIF1α AD-synthetic TF had a G/V ratio of 0.36 on a day the aggregate G/V ratio for SNAP-tag2 alone was 0.26, its normalized G/V ratio is 1.4.

Finally, we calculated the mean and derived error bounds for the normalized PAPA ratios by bootstrapping. For a condition comprising n cells (imaged over multiple days, but each normalized to their appropriate SNAP-tag2 control), we picked n cells from that condition with replacement and calculated the combined normalized G/V ratio of this pool. We repeated this 96 times and used this to derive 95% confidence intervals.

## FRAP movie acquisition

The day of imaging, cells were stained for 5 min with TMR-HaloTag ligand (Promega) added to a final concentration of 50 nM, rinsed once with imaging medium, and incubated in imaging medium for 10 min. The dishes were rinsed one more time, replenished with imaging medium, and mounted on the microscope with Immersol 518 F (Zeiss). The dish was then left to equilibrate thermally for about 5 min prior to imaging.

FRAP was conducted using a ZEISS LSM900 Airyscan 2 laser-scanning confocal microscope mounted on an inverted Axio Observer.Z1/7 platform and operated with ZEN 3.1 blue software. The system was equipped with a temperature- and $CO_2$-controlled incubation chamber (Zeiss/PeCon) maintained at 37°C and 5% $CO_2$. Fluorophores were excited using a 561 nm laser with a maximally sized pinhole, 3.5% laser power, and 800 V detector gain. Movies were acquired with a 40× oil-immersion objective with a numerical aperture of 1.3 using bidirectional scanning to maximize scan speed. Intentional photobleaching (100% 561 nm laser and 100% 405 nm laser) was applied during frame 16.

## FRAP analysis

We used custom Python code (https://github.com/vinsfan368/FRAPpy, copy archived at *Fan, 2025b*) to quality-control FRAP movies and calculate normalized recoveries. FRAP movies were rejected if there was substantial axial drift.

Fitting FRAP models to experimental data carries model-dependent assumptions (*Mueller et al., 2008*; *Sprague et al., 2004*). We followed the analysis scheme outlined by *Mueller et al., 2012*, subheading 3.3.2. Briefly, this involves:

1. Defining a nuclear mask for each FRAP movie. We took a sum-intensity projection across the FRAP movie after normalizing each frame by mean subtraction and dividing by each frame's standard deviation. We applied a Gaussian blur to this image and used isodata thresholding followed by hole-filling to define the nucleus.
2. Background subtracting each frame of the FRAP movie. We took the median value of the pixels of the frame defined as non-nuclear as the 'background' of that frame and subtracted this value from all of the pixels in that frame.
3. Normalizing fluorescence values within the nuclear mask to correct for observational photobleaching. Images in the pre-bleach phase of the movie were normalized such that their nuclear intensities were equal to that of the last frame before photobleaching. Images in the post-bleach phase of the movie were normalized such that their nuclear intensities were equal to that of the first image acquired after bleaching.
4. Calculating FRAP(t), the fluorescence recovery curve over time. At each time point, this was defined as the mean FRAP spot intensity. (These values have been background-subtracted in step 2 and corrected for observational photobleaching in step 3.) We then divided these values by the mean of the pre-bleach frames, which normalizes the FRAP(t) curve to 1.
5. Resampling FRAP(t) using log-spaced bins to prevent overrepresentation of the long FRAP recovery tail. We defined 100 log-spaced bins between 0.01 s and the max time point recorded for any of the movies in a condition. Then, we took the arithmetic mean of both the timestamps and normalized recovery values falling between these bins. These were used to fit recovery curves below.
6. Testing if diffusion plays a role in FRAP recovery with the gradient-smoothing test. We plotted the radial intensity profile of the frames immediately after bleaching, normalized them to between 0 and 1, and checked for shape changes as recovery progressed. We did not see changes that would indicate diffusion is contributing to our measured FRAP recoveries for any condition (*Figure 2—figure supplement 4D-F*). We also note that for the radius of our bleach

spot (0.75 μm) and the diffusion coefficient measured by SMT (slower-diffusing population: ~2 μm²/s), we would expect diffusive molecules on the edge of the bleach circle to reach the center in about 70 ms on average (mean squared displacement = 4Dt), and our imaging setup allowed us to capture the first post-bleach frame only ~250 ms after bleaching.

7. Fitting the FRAP(t) curves to a single-exponential recovery function:

$$\mathrm{FRAP}(\Delta t) = A\left(1 - e^{-\Delta t/\tau}\right)$$

with an implicit constant term constraining A to [0, 1]. Only FRAP recovery for H2B-HaloTag was well fit by this equation. We instead fit FRAP recoveries for the synthetic TFs to a double-exponential recovery function:

$$\mathrm{FRAP}(\Delta t) = A_1\left(1 - e^{-\Delta t/\tau_1}\right) + A_2\left(1 - e^{-\Delta t/\tau_2}\right)$$

also with an implicit constant term constraining $A_1$ and $A_2$ to [0, 1] and their sum to [0, 1].

We have tabulated the values from these fits in *Supplementary file 1*.

## Cleavage under targets and release using nuclease (CUT&RUN) protocol

We followed steps 1–13, 21–37, and 57–60 of a published CUT&RUN protocol (https://doi.org/10.17504/protocols.io.zcpf2vn), a revised protocol from *Meers et al., 2019*, with the following modifications.

Cells were counted prior to mixing with concanavalin A beads, and ~299,000 cells were used for each condition. All cell centrifugation steps were 500×*g* for 3 min. The digitonin concentration optimized for our U2OS cell lines was 0.02%. For all experimental conditions, we used a polyclonal rabbit anti-V5 antibody (Abcam ab9116). The wash buffer and Dig-wash buffer were supplemented with 1 μM estradiol. All wash steps with the Dig-wash buffer were repeated once (for a total of two washes) with 200 μL of Dig-wash buffer.

Following phenol-chloroform extraction, we used ferromagnetic bead cleanup to concentrate and purify DNA fragments:

1. Add 200 μL SPRI-like homemade magnetic beads to 200 μL eluate from the phenol-chloroform extraction.
2. Add 400 μL of PEG-8000 buffer to this mixture to help precipitate the DNA fragments.
3. Rotate at room temperature for 1 hr.
4. Place the tubes on a magnet stand.
5. Pipette off supernatant.
6. Add 800 μL of freshly made 80% ethanol to the tube, being careful not to disturb the beads.
7. Pipette off supernatant.
8. Repeat steps 6 and 7.
9. Air-dry the tubes until beads appear dull brown but not dry and cracked.
10. Elute in 21 μL $H_2O$.

For library preparation, we used NEBNext Ultra II DNA Library Prep Kit for Illumina and followed the manufacturer's instructions with the following modifications.

The Illumina hairpin adapter was diluted 25-fold (i.e. 1 part concentrated adapter plus 24 parts water). Eleven PCR cycles were used for amplification. We performed a dual-sided bead selection with homemade SPRI-like beads with bead concentrations 0.67× (and discarding bead-bound large fragments) and 1.0× (and keeping bead-bound fragments). Fragment sizes were checked by Bioanalyzer (Agilent). Libraries were pooled replicate-wise and sequenced on a NextSeq 2000. Two replicates for each condition were sequenced.

## Genomics analysis

We generated a custom hg38 genome with our reporter locus engineered at the AAVS1 locus with bespoke bash scripts. We generated bowtie2 indices for this genome and used this as a reference for

both ATAC-seq and CUT&RUN data. We also used the blacklist from *Nordin et al., 2023*, for both ATAC-seq and CUT&RUN analyses.

We used publicly available ATAC-seq data (*Oomen et al., 2019*) in U2OS cells as a baseline for genomic accessibility in our cell line. We used the nf-core (*Ewels et al., 2020*) pipeline atacseq (*Patel et al., 2023*) to generate genomic accessibility signal (bigWig files) and to call peaks (BED files). We used the pipeline with the following non-default parameters: aligner: 'bowtie2', narrow_peak: true, min_reps_consensus: 2. This dataset had two biological replicates with four technical replicates each; we merged peak sets between the two biological replicates with custom bash scripts for our analyses. For genomic tracks, we took the mean of the two bigWig files corresponding to the two biological replicates.

For our CUT&RUN data, we used the cutandrun (*Cheshire et al., 2024*) nf-core pipeline with the following non-default parameters: spikein_genome: BDGP6, trim_nextseq: 20, remove_mito-chondrial_reads: true, mito_name: chrM, dedup_target_reads: true, use_control: false, seacr_peak_threshold: 0.01, replicate_threshold: 2. This normalized our occupancy data to sheared *Drosophila* DNA introduced in the CUT&RUN STOP buffer. For genomic tracks and quantification of signal, we took the mean of the two bigWig files corresponding to two biological replicates for each of cell lines expressing synthetic TFs. For peak sets, we limited ourselves to peaks that were reproduced in both replicates. To call CUT&RUN peak overlaps with ATAC-seq peaks (*Figure 7—figure supplement 1A*), we considered a peak overlapping if at least one base pair of the CUT&RUN peak was shared with any ATAC-seq peak.

## Materials availability

Plasmids used in this study have been deposited with Addgene (Addgene ID numbers 240165–240174). Cell lines are available upon reasonable request.

## Acknowledgements

A heartfelt thank you to all members of the Tjian+Darzacq and Staller groups for their intellectual input and experimental advice for this project, particularly Djem Kissiov, Claudia Cattoglio, Joseph McKenna, Yu Chen, Aditya Udupa, Oscar Whitney, and Alec Heckert. Thank you to Anna Mapp for suggesting the norstictic acid experiment. A large thanks to Sathvik Anantakrishnan for his efforts on microscope automation and maintenance, without which this project would not have been possible. We thank Djem Kissiov, Aditya Udupa, and Robert Tjian for the careful reading of the manuscript. We thank Anders Näär and Haribabu Arthanari for helpful discussions. We thank Luke Lavis for providing us with Janelia Fluor dyes. We thank James McNally for helpful correspondence and code sharing. We thank the Dillin lab, particularly Kimberly Tsui, for training and use of the Incucyte S3 instrument.

## Additional information

### Competing interests

Xavier Darzacq: is a co-founder of Eikon Therapeutics, Inc. The other authors declare that no competing interests exist.

### Funding

| Funder | Grant reference number | Author |
|---|---|---|
| Simons Foundation | 1018719 | Max V Staller |
| National Institute of General Medical Sciences | R35GM150813 | Max V Staller |
| National Science Foundation | 2112057 | Max V Staller |
| National Institute of General Medical Sciences | 1RM1GM139738 | Thomas GW Graham |

| Funder | Grant reference number | Author |
|---|---|---|
| Silicon Valley Community Foundation | RR-8175 | Xavier Darzacq |
| Biohub, San Francisco | | Max V Staller |

The funders had no role in study design, data collection and interpretation, or the decision to submit the work for publication.

## Author contributions

Vinson B Fan, Conceptualization, Data curation, Formal analysis, Investigation, Visualization, Writing – original draft, Writing – review and editing; Abrar A Abidi, Methodology; Thomas GW Graham, Methodology, Writing – review and editing; Xavier Darzacq, Supervision, Funding acquisition, Writing – review and editing; Max V Staller, Conceptualization, Supervision, Writing – original draft, Project administration, Writing – review and editing

## Author ORCIDs

Vinson B Fan ⓘ https://orcid.org/0000-0002-1688-7780
Xavier Darzacq ⓘ https://orcid.org/0000-0003-2537-8395
Max V Staller ⓘ https://orcid.org/0000-0001-9094-5697

## Decision letter and Author response

Decision letter https://doi.org/10.7554/eLife.105776.sa1
Author response https://doi.org/10.7554/eLife.105776.sa2

---

# Additional files

## Supplementary files

Supplementary file 1. Fluorescence recovery after photobleaching (FRAP) parameters extracted from a double-exponential recovery model. FRAP population fractions and recovery times for cells expressing various synthetic transcription factors (TFs) and H2B. Stronger activation domains have longer slow-recovery times, indicating longer residence times bound to chromatin. *Note that >95% of H2B molecules do not recover on the timescale of the FRAP experiment.

Supplementary file 2. Summary of single-molecule tracking (SMT) measurements. Fraction bound, number of cells imaged across all replicates, number of trajectories, number of jumps, and number of detections for all SMT experiments reported in this article. All tabulated values are post-filtering (Materials and methods: SMT and PAPA-SMT analysis). Number of trajectories and number of detections include singlets: detections that were not linked to any other detection and do not contribute any data to downstream analyses but have their own 'trajectory' index. * indicates datasets acquired without an automated nucleus-finding macro.

MDAR checklist

## Data availability

SMT trajectories (CSV files tabulating all detections surviving filtering steps) are available on Dryad (https://doi.org/10.5061/dryad.41ns1rnqt). We were unable to deposit underlying SMT movies due to their large size. The raw movies are available on request for non-commercial use. Please email the corresponding author to make data transfer arrangements. A representative subset of the raw data and a processed version of the dataset have been deposited in Dryad. FRAP movies are also available as part of the same Dryad dataset (https://doi.org/10.5061/dryad.41ns1rnqt). CUT&RUN data has been deposited to the Sequence Read Archive under Accession PRJNA1305492. ATAC-seq data from *Oomen et al., 2019* is available on the GEO as Series GSE121840. Plasmids have been submitted to Addgene and are available under IDs 240165–240174. All essential code is available on public GitHub repositories. We used quot to detect, localize, and connect spots in SMT movies; commit 1b9051e of (https://github.com/vinsfan368/quot, copy archived at *Heckert, 2025b*) was used for this article. We used qtpicker to refine nuclear masks; commit 6bcbf12 of (https://github.com/vinsfan368/qtpicker, copy archived at *Fan, 2025a*) was used for this article. We used saspt (*Heckert et al., 2022*) to generate posterior diffusion spectra; commit 632faad of (https://github.com/vinsfan368/saspt, copy archived at *Fan, 2025c*) was used for this article. We used FRAPpy to analyze FRAP movies from CZI

files; commit dc3687d of (https://github.com/vinsfan368/FRAPpy, copy archived at *Fan, 2025b*) was used for this article.

The following datasets were generated:

| Author(s) | Year | Dataset title | Dataset URL | Database and Identifier |
|---|---|---|---|---|
| Fan VB | 2025 | Data from: Short activation domains control chromatin association of transcription factors | https://doi.org/10.5061/dryad.41ns1rnqt | Dryad Digital Repository, 10.5061/dryad.41ns1rnqt |
| Fan VB | 2025 | Short activation domains control chromatin association of transcription factors | https://www.ncbi.nlm.nih.gov/bioproject/?term=PRJNA1305492 | NCBI BioProject, PRJNA1305492 |

The following previously published dataset was used:

| Author(s) | Year | Dataset title | Dataset URL | Database and Identifier |
|---|---|---|---|---|
| Oomen ME, Hansen AS, Liu Y, Darzacq X, Dekker J | 2018 | CTCF sites display cell cycle dependent dynamics in factor binding and nucleosome positioning | https://www.ncbi.nlm.nih.gov/geo/query/acc.cgi?acc=GSE121840 | NCBI Gene Expression Omnibus, GSE121840 |

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
