## [Editor Report]

This useful study presents evidence that the duration and/or the frequency with which transcription factors interact with chromatin/DNA in living cells is influenced by the transactivation domains of transcription factors. The methods used are solid, combining imaging and genomics experiments. The work will be of interest to molecular biologists and biochemists, working in the transcriptional regulation field.

---

## [Decision Letter]

**Decision letter after peer review:**

Thank you for submitting your article "Short activation domains control chromatin association of transcription factors" for consideration by *eLife*. Your article has been reviewed by 2 peer reviewers, and the evaluation has been overseen by a Reviewing Editor and Volker Dötsch as the Senior Editor. The reviewers have opted to remain anonymous.

Essential Revisions:

1) Add, change and discuss the literature suggested by reviewer #1 in points 1-4.

2) This work clearly demonstrates that stronger TADs result in stronger/more frequent TF binding 'genome-wide'. However the data presented in this manuscript does not allow to conclude if such stronger binding has some specific effects at the binding sites, or if stronger regulation and stronger genomewide binding are two independent events. Performing chromatin immunoprecipitation(or single molecule footprinting, as in Doughty et al., 2024) experiments at the binding sites of the reporter gene, to test the occupancy of the same DBD with different TADs would significantly increase the impact of this work.

3) The authors should cite some previous papers that, by using SMT, have tested the effect of removing TADs on endogeneous TF binding and diffusion. Some of them (Chen et al., Cell 2014) have shown that less binding to chromatin is observed, in agreement with this study, while others (Callegari et al., Embo Rep, 2019, Mazzocca et al., Nat Comm., 2023), have shown no effect on binding, but a modulation of their diffusion properties.

4) The authors present an interesting methodological approach to extract errors from SMT data, using cell-based boot-strapping. This allows to provide estimates of Bound fractions with error bars (Figure 3D). The authors should include error estimates using the same approach also in Figure 2C.

*Reviewer #1 (Recommendations for the authors):*

The authors have already shown that sequences outside the HIF DBD can affect chromatin occupancy; the present paper refines this to activation domains. Activation domains are well known to directly interact with nucleosome remodelers and histone acetylase complexes and recruit them to the genome. Such interactions/recruitment are independent of specific DNA-binding and would occur throughout the genome, albeit at much lower levels for a non-specific sequence. However, non-specific sequences far outnumber specific target sites, and assays of total genome association represent contributions from both. Nucleosome remodelers and histone acetylase complexes typically (and maybe always) strongly bind to DNA and/or histones. Thus, it is obvious that the combination of specific DNA-binding domains and associated chromatin-modifying activities will result in higher genome association and residence times. The paper is useful in that it provides experimental evidence for the obvious, but it doesn't break any new ground.

The paper hypes the results by seriously misrepresenting the "traditional view". I don't think anyone ever claimed the so-called traditional view that "the DBD is solely responsible for binding chromatin". Instead, the DBD was historically viewed as being solely responsible for binding to specific target sites, although this has been challenged recently by work from the Barkai lab. The Barkai work indicates that sequences outside the DBD can affect target specificity, but this is not due to activation domains. To put it differently, traditional views of transcription were not concerned with generic chromatin association especially given that activation domains recruit chromatin-modifying activities that interact with nucleosomes and DNA.

The scholarship is weak.

1. First, Brent and Ptashne 1985 was an important paper, but it only showed that DNA-binding domains of activators could be swapped. It did not map an activation domain. That was first done by Hope and Struhl 1986 and elaborated upon by subsequent papers from the Ptashne and Struhl labs (Gill and Ptashne, 1987 Cell; Ma and Ptashne, 1987 Cell; Hope et al., 1988 Nature; see also reviews Struhl, 1987 Cell; 1989 Ann Rev. Biochem; Ptashne 1988 Nature; Ptashne 1989 Sci Amer).

2. Brent and Ptashne 1985 never proposed that "the DBD scans the DNA and binds to a cognate motif and only then does the activation domain recruit coactivators from solution". That highly mechanistic statement is far beyond anything Brent and Ptashne said, and it should be noted that coactivators weren't even known in 1985. To put it different, all the early papers clearly distinguished between specific DNA-binding and transcriptional activation domains, but they never addressed generic chromatin binding or kinetic mechanisms.

3. It is strange that the authors uniquely cite Gill et al. 1994 for activation domains interact with and recruit the basal transcription machinery including Pol II. That paper showed an in vitro interaction between the Sp1 glutamine-rich activation domain and TAF110, but the paper claiming physiological relevance (Sauer et al. 1996) was fraudulent and retracted. On the other hand, it is surprising that the authors do not cite any papers related to Mediator, which is a well-established target of numerous activation domains or more recent work from Roeder showing that TFIID can be a target in selected cases.

4. Ning et al. 2022 and Sun et al. 2021 are interesting papers, but they have nothing to do with challenges to the "traditional view". There are decades of examples of intramolecular interactions that affect biochemical activities of proteins.

The experiments and results per se are fine, but the context in which they are placed is highly misleading and this needs to be significantly modified.

*Reviewer #2 (Recommendations for the authors):*

In this work Fan, Darzacq and Staller, perform live-cell single molecule tracking (SMT) on a collection of synthetic transcription factors (sTFs), previously developed by the last author, and correlate the dynamics sTFs with the expression of a reporter gene. The authors find that – contrarily to the standard model for transcriptional regulation – association to chromatin does not only depend on the DNA binding domain. Rather, sTFs with activation domains that result in stronger activity, also display slower dynamics, possibly representing tighter (or more frequent) associations with chromatin. These results were further corroborated by looking at single molecule dynamics of two different natural DNA binding domains fused with activation domains of different strength. Finally perturbation experiments using a couple of small-molecule inhibitors were used to show that interaction of transcriptional coactivators with TFs activator domains might be responsible for increasing TF dwell time on chromatin.

Although some hints on the role of activation domains on transcription factor binding could be found in other SMT studies from other groups, this work has the strength of resorting to artificial, tuneable systems to show that this phenomenon might be shared among multiple classes of transcription factors.

The conclusions of the paper are – to a large extent – supported by the presented data, except for a couple of claims that could benefit from some more extended analysis of the data.

1) The authors claim that the constructs with stronger activation domains display higher bound fractions and longer residence times on chromatin. The fraction of bound molecules is measured by calculating the spectra of diffusion coefficients and by kinetic modelling of single molecule displacements, approaches that have been validated by multiple studies. The claim about residence times is instead based on Fluorescence Recovery after Photobleaching experiments (FRAP, Figure 2D, Figure S4B-C, Figure S6D), and this analysis is a bit more problematic. Indeed, as the authors correctly report, multiple phenomena and multiple models can explain FRAP recovery in presence of a mixture of diffusion and binding (Sprague, Biophys J., 2004, Mueller, Curr. Op. Cell Biol., 2010), and slower recovery curves can be caused by either slower dissociation (longer residence times), faster association, or slower diffusion – or a combination of these. Identifying whether longer binding or more frequent association to chromatin would be important, especially considering that recent SMF data is highlighting that also in mammalian cells on-rates are more relevant than off-rates in controlling occupancy of binding sites (Schaepe et al.,, BiorXiv, 2025). One possibility to address these issues would be to fit the FRAP data with the kinetic models that estimate binding rates (Mueller et al., Biophys J, 2008). An alternative solution could be to resort to slow frame-rates single molecule tracking to retrieve the distribution of binding times, and then estimate off-rates with the one of the approaches previously developed for this goal (Chen, Cell, 2013, Loffreda, Nat. Comm. 2017, Hansen, *eLife*, 2017, Nguyen, Mol Cell, 2021).

2) The discussion of the results on the p300 inhibitor (Figure 5A, incorrectly mislabeled as Figure 5B in the figure legend) was not very clear for me. The authors show that a drug that stabilize p300 binding on chromatin, also stabilize the binding of a sTF. Yet, this could be an indirect effect, and does not directly demonstrate that TFs can be tethered to chromatin through interactions with p300. At a minimum the effects of the p300 inhibitor on the chromatin interactions of same DBD, but without an activator domain, should be shown to support the authors' claim. A similar control would help discussing the results on the BAF inhibitor (Figure 5B, incorrectly mislabeled as Figure 4A in the figure legend).

3) This work clearly demonstrates that stronger TADs result in stronger/more frequent TF binding 'genome-wide'. However the data presented in this manuscript does not allow to conclude if such stronger binding has some specific effects at the binding sites, or if stronger regulation and stronger genomewide binding are two independent events. Performing chromatin immunoprecipitation(or single molecule footprinting, as in Doughty et al., 2024) experiments at the binding sites of the reporter gene, to test the occupancy of the same DBD with different TADs would significantly increase the impact of this work.

4) The authors should cite some previous papers that, by using SMT, have tested the effect of removing TADs on endogeneous TF binding and diffusion. Some of them (Chen et al., Cell 2014) have shown that less binding to chromatin is observed, in agreement with this study, while others (Callegari et al., Embo Rep, 2019, Mazzocca et al., Nat Comm., 2023), have shown no effect on binding, but a modulation of their diffusion properties.

5) The authors present an interesting methodological approach to extract errors from SMT data, using cell-based boot-strapping. This allows to provide estimates of Bound fractions with error bars (Figure 3D). The authors should include error estimates using the same approach also in Figure 2C.

[Editors' note: further revisions were suggested prior to acceptance, as described below.]

Thank you for resubmitting your work entitled "Short activation domains control chromatin association of transcription factors" for further consideration by *eLife*. Your revised article has been evaluated by Volker Dötsch (Senior Editor).

Please address the concerns of reviewer 1 that are listed below.

*Reviewer #1 (Recommendations for the authors):*

Although the revised paper is improved by better literature citation and a new experiment, I'm sorry to say that my overall opinion hasn't changed. As I stated before, the experiments are useful, but the main conclusions are incremental, entirely expected, and hardly challenge traditional views of anything. I don't have a problem with the paper being published in *eLife*, but I recommend rejection if the authors refused to remove all statements claiming that their paper challenges traditional/previous views.

1. As can be seen from the figure and associated text in their response letter, the authors have misunderstood my main point. This authors' figure concerns cooperative effects involving separate DNA-binding proteins bound to separate sequences. That can certainly happen, but it has nothing to do with my main point, and it has nothing to do with their observations. Instead, my point is that activation domains directly interact with chromatin-modifying activities (e.g. nucleosome remodelers such as Swi/Snf and histone acetylases such as SAGA/Gcn5) that directly interact with DNA and/or histones. As such, the cooperativity that causes increased chromatin association does not involve a second DNA-binding but rather occurs between a single DNA-binding protein and the recruited chromatin-modifying activity. All DNA-binding proteins have low-level affinity for all sequences (typically called non-specific binding), and the interaction of activation domains and chromatin-modifying activities is independent of specific DNA-binding and would occur throughout the genome, albeit at much lower levels for a non-specific sequence. Thus, it is obvious that the combination of specific DNA-binding domains and associated chromatin-modifying activities will result in higher genome association and residence times. The paper is useful in that it provides experimental evidence and measurements for this, but it doesn't break any new ground, and it certainly doesn't challenge traditional views.

2. Although the citations are significantly improved, the paper still hypes the results by claiming to overturn the "traditional view". No one has ever claimed the so-called traditional view that "the DBD is solely responsible for binding chromatin". As such, the results here cannot challenge a view that no one has expressed or even believed. Instead, the authors have invented this "traditional view" and used it as a straw man to shoot down. All the early work on activation domains and the vast majority of subsequent work ignored generic chromatin association because it was hard to imagine how it would contribute to the specificity of gene regulation. Even with this paper, the biological significance of generic chromatin association is unclear. I do agree that the measurements here are useful, but this is far from overturning a traditional view that has not even been expressed.

3. Regarding the key concept of modularity of DNA-binding and activation domains, the authors have invented ideas that were not expressed or even contemplated in previous papers. The original papers on modularity were concerned with specific DNA-binding and transcriptional activation. They were published well before the concept of co-activators, and indeed the prevailing view based on bacterial activators is that activation domains would contact some component of the basic Pol II machinery. Indeed, there were numerous experiments claiming that various general factors (e.g. TBP, TFIIB, TFIIA) were the target. These early papers establishing modularity were also well before any connection to chromatin, and at the time, chromatin was largely ignored in the gene expression field. Eventually, it became clear that activation domains not only stimulated transcription directly (mostly through Mediator) but also indirectly through chromatin via Swi/Snf and SAGA. The connection of activation domains and chromatin is 30 years old, so using early papers on modularity to claim that activation domains had nothing to do with chromatin is historically unfair. And, this 30-year connection between activation domains and chromatin-modifying activities is what renders the results in this paper entirely predictable.

*Reviewer #2 (Recommendations for the authors):*

The authors have satisfactorily addressed my comments.

In particular, the addition of CUT&RUN experiments clarifies that swapping ADs only slightly impacts where the TF will bind, but rather modulates binding frequency and/or stability at pre-determined sites. I think that this result should be highlighted at the end of the introduction. Also, as a minor point I would be curious to see if the motifs bound by the different TFs variants are identical, an observation that could further support their observation.

---

## [Author Response]

Essential Revisions:1) Add, change and discuss the literature suggested by reviewer #1 in points 1-4.

Thank you for these suggestions. We have completely rewritten the introduction to discuss these four points.

2) This work clearly demonstrates that stronger TADs result in stronger/more frequent TF binding 'genome-wide'. However the data presented in this manuscript does not allow to conclude if such stronger binding has some specific effects at the binding sites, or if stronger regulation and stronger genomewide binding are two independent events. Performing chromatin immunoprecipitation(or single molecule footprinting, as in Doughty et al., 2024) experiments at the binding sites of the reporter gene, to test the occupancy of the same DBD with different TADs would significantly increase the impact of this work.

We have undertaken CUT&RUN on 5 synthetic TFs. These results are included in the new Figure 7. As the reviewer notes, there are two primary possibilities: TFs with a higher bound fraction could bind the same genomic loci with higher occupancy, or they could bind new loci.

Looking at the reporter locus, we see binding by all five TFs, including the ‘no activation domain’ control. At the reporter, the stronger activation domains have higher occupancy.

Comparing the HIF1α WT and superactive alleles show they have overlapping genome-wide binding patterns. The superactive allele binds the same sites as WT but more strongly, and it binds additional sites. The majority of all sites bound by both TFs are preexisting ATAC-seq peaks.

Genome-wide, we find that the majority of TF binding overlaps with existing ATAC-seq peaks. We conclude that the primary effect observed in the SMT is increased binding at the active loci.

3) The authors should cite some previous papers that, by using SMT, have tested the effect of removing TADs on endogeneous TF binding and diffusion. Some of them (Chen et al., Cell 2014) have shown that less binding to chromatin is observed, in agreement with this study, while others (Callegari et al., Embo Rep, 2019, Mazzocca et al., Nat Comm., 2023), have shown no effect on binding, but a modulation of their diffusion properties.

These are fantastic suggestions. We have incorporated these ideas and citations. Please see pg 3 line 92-93 and pg 22 line 505.

4) The authors present an interesting methodological approach to extract errors from SMT data, using cell-based boot-strapping. This allows to provide estimates of Bound fractions with error bars (Figure 3D). The authors should include error estimates using the same approach also in Figure 2C.

We have added error bars to Figure 2C.

Reviewer #1 (Recommendations for the authors):The authors have already shown that sequences outside the HIF DBD can affect chromatin occupancy; the present paper refines this to activation domains. Activation domains are well known to directly interact with nucleosome remodelers and histone acetylase complexes and recruit them to the genome. Such interactions/recruitment are independent of specific DNA-binding and would occur throughout the genome, albeit at much lower levels for a non-specific sequence. However, non-specific sequences far outnumber specific target sites, and assays of total genome association represent contributions from both. Nucleosome remodelers and histone acetylase complexes typically (and maybe always) strongly bind to DNA and/or histones. Thus, it is obvious that the combination of specific DNA-binding domains and associated chromatin-modifying activities will result in higher genome association and residence times. The paper is useful in that it provides experimental evidence for the obvious, but it doesn't break any new ground.

Although it is always possible that interactions between two proteins can facilitate cooperative binding to chromatin, it is not obvious that this cooperative binding will occur to an appreciable extent at physiologically relevant concentrations in live cells. Critically, the magnitude of the cooperative binding cannot be predicted *a priori*. It must be measured.

Consider Author response image 1:

**Author response image 1. sa2fig1:** Conceptual model for how a coactivator can lead to apparent cooperativity between transcription factors. (**A**) A transcription factor nonspecifically-bound to random DNA. (**B**) A transcription factor specifically-bound to a cognate DNA motif at our reporter. (**C**) A transcription factor non-specifically-bound to random DNA and simultaneously bound to a coactivator. The coactivator is also bound to another transcription factor specifically bound to its cognate motif. This situation represents other active loci. (**D**) A transcription factor specifically-bound to a cognate DNA motif and simultaneously bound to a coactivator tethered to the DNA by another transcription factor. This binding mode could occur at our reporter locus, but the identity of the dark blue transcription factor is unknown.

Specific-binding to a motif (B) will have a longer lifetime on DNA and higher bound fraction than non-specific binding to random DNA (A). Simultaneous specific binding to a DNA motif and a coactivator (D) will have a longer lifetime than non-specific binding to random DNA and a coactivator (C). In addition, D will have more stable binding than B because a doubly-bound protein requires two dissociation events before floating away.

In contrast, it is difficult to predict *a priori* if C will have a longer dwell time than B, and it will depend on the relative affinities of the three interactions.

We agree that multivalent binding will increase dwell time on DNA. However, we were surprised by the magnitude of the difference. We expected C and D to be very rare compared to A, such that they would not be detectable in the SMT data.

Our direct live-cell measurements show that activation domains exert a strikingly large effect on chromatin binding of synthetic TFs. A DBD fused to the superactive K>A mutant of the CITED2 AD had a bound fraction of 64%, for instance, while the same DBD fused to the inactive L>F mutant had a bound fraction of only 25%. In this case, the AD contributes the majority of binding. This large magnitude was unexpected and not obvious.

The magnitude of this effect is all the more striking given that—as the reviewer correctly notes—many TF binding events are thought to occur outside of regulatory elements on "nonspecific" DNA. One might have reasonably predicted that only a small fraction of TF ADs—perhaps a few percent co-bound with other TFs at regulatory elements—engage coactivators at any instant, such that these interactions would have little effect on overall chromatin binding measured by SMT. Alternatively, one might have reasonably predicted that AD-coactivator interactions would be too weak and transient to impact global chromatin binding. While previous results showed that long IDRs can influence both overall chromatin binding and binding specificity of TFs, it was unclear whether activation domains *per se* were responsible or whether minimal ADs would have the same effect.

Instead, our results imply that at any instant, a large fraction of TF ADs contact other proteins—likely coactivators—such that ADs exert a large effect on overall chromatin binding of TFs. This result was not obvious, and such a claim would have been mere speculation prior to our work.

The paper hypes the results by seriously misrepresenting the "traditional view". I don't think anyone ever claimed the so-called traditional view that "the DBD is solely responsible for binding chromatin". Instead, the DBD was historically viewed as being solely responsible for binding to specific target sites, although this has been challenged recently by work from the Barkai lab. The Barkai work indicates that sequences outside the DBD can affect target specificity, but this is not due to activation domains. To put it differently, traditional views of transcription were not concerned with generic chromatin association especially given that activation domains recruit chromatin-modifying activities that interact with nucleosomes and DNA.

We agree with the reviewer that "the DBD was historically viewed as being solely responsible for binding to specific target sites." The reviewer also articulates what we consider the traditional view, here and above, namely that the function of ADs is to "recruit" other proteins to chromatin.

Our key point is that this traditional perspective is incomplete. TFs do not bind naked DNA in cells; they bind chromatin alongside many other proteins. Activation domains do not just recruit coactivators; they and coactivators recruit each other. Interactions between TFs, coactivators, and chromatin must be understood collectively, not in isolation, which requires probing these interactions under physiological conditions in live cells. We have done this systematically for a series of activation domains, and we believe that our results are not over-hyped.

Our results have several important implications. First, the strong effect of activation domains on chromatin binding implies that a substantial fraction (~40% or more) of activation domains are bound by coactivators in live cells. To our knowledge, this fraction has not previously been measured, and it provides an important quantitative constraint for models of transcription. Second, the strong global effect of activation domains on chromatin binding suggests that activation domains could likewise influence binding of TFs at specific sites—dependent on local coactivator enrichment—contrary to the historical view that DBDs are "solely responsible." Third, as noted by the reviewer, our results extend previous work from the Barkai Lab and others: we demonstrate that short activation domains, isolated from longer IDRs, are sufficient to affect the overall affinity of TFs for chromatin.

Separately, we wish to point out that there are many review articles and textbooks that expose the idea that the “DBD is solely responsible for binding DNA.” We suspect that Ptashne pushed TF modularity and his recruitment model so strongly that more nuanced ideas were lost amongst his shouting.

“Activators are modular proteins whose activating regions are readily interchangeable; so are their DNA-binding regions” –Ptashne M. 1988. How eukaryotic transcriptional activators work. *Nature*
**335**:683–689.

“It is believed that the DNA binding domain serves merely to bring the protein to the DNA target, whereupon the activation region stimulates the basic transcription machinery” –Struhl K. 1993. Yeast transcription factors. *Curr Opin Cell Biol* 5:513–520.

“Site-specific TFs are modular in their structure reflecting their ability to bind to DNA via their DNA binding domains and simultaneously bind to other transcriptional regulatory proteins via so-called effector domains. The modular nature of site-specific TFs has been repeatedly demonstrated using in vitro and in vivo reporter assays.” –Frietze S, Farnham PJ. 2011. Transcription factor effector domains. *Subcell Biochem*
**52**:261–277.

“Most gene regulatory proteins that activate gene transcription—that is, most gene activator proteins—have a modular design consisting of at least two distinct domains. One domain usually contains one of the structural motifs discussed previously that recognizes a specific regulatory DNA sequence. In the simplest cases, a second domain—sometimes called an activation domain—accelerates the rate of transcriptional initiation.” –Alberts et al. *The Molecular Biology of the Cell* 4th edition, 2004, pg 401

“Extensive studies on a variety of transcription factors have shown that they have a modular structure in which distinct regions of the protein mediate particular functions such as DNA binding (see Chapter 4) or interaction with specific effector molecules such as steroid hormones. It is likely therefore that a specific region of each individual transcription factor will be involved in its ability to activate transcription following DNA binding. As described in Chapter 2 (section 2.4.1), such activation domains have been identified by so-called 'domain swap' experiments in which various regions of one factor are linked to the DNA binding domain of another factor and the ability to activate transcription assessed.

In general, these experiments have confirmed the modular nature of transcription factors with distinct domains mediating DNA binding and transcriptional activation.” –Latchman Eukaryotic Transcription Factors 5th edition, 2008, pg 161.

“Proteins that regulate transcription in eukaryotes are typically modular in nature, with a DNA-binding domain that recognizes specific DNA sequences and a separate activation or repression domain that recruits a co-activator or co-repressor, respectively” –Craig, Green, Storz et al. *Molecular Biology: Principles of Genome Function* 3rd edition, 2021, pg 344.;

The scholarship is weak.1. First, Brent and Ptashne 1985 was an important paper, but it only showed that DNA-binding domains of activators could be swapped. It did not map an activation domain. That was first done by Hope and Struhl 1986 and elaborated upon by subsequent papers from the Ptashne and Struhl labs (Gill and Ptashne, 1987 Cell; Ma and Ptashne, 1987 Cell; Hope et al., 1988 Nature; see also reviews Struhl, 1987 Cell; 1989 Ann Rev. Biochem; Ptashne 1988 Nature; Ptashne 1989 Sci Amer).2. Brent and Ptashne 1985 never proposed that "the DBD scans the DNA and binds to a cognate motif and only then does the activation domain recruit coactivators from solution". That highly mechanistic statement is far beyond anything Brent and Ptashne said, and it should be noted that coactivators weren't even known in 1985. To put it different, all the early papers clearly distinguished between specific DNA-binding and transcriptional activation domains, but they never addressed generic chromatin binding or kinetic mechanisms.3. It is strange that the authors uniquely cite Gill et al. 1994 for activation domains interact with and recruit the basal transcription machinery including Pol II. That paper showed an in vitro interaction between the Sp1 glutamine-rich activation domain and TAF110, but the paper claiming physiological relevance (Sauer et al. 1996) was fraudulent and retracted. On the other hand, it is surprising that the authors do not cite any papers related to Mediator, which is a well-established target of numerous activation domains or more recent work from Roeder showing that TFIID can be a target in selected cases.4. Ning et al. 2022 and Sun et al. 2021 are interesting papers, but they have nothing to do with challenges to the "traditional view". There are decades of examples of intramolecular interactions that affect biochemical activities of proteins.

We have completely rewritten the introduction to address these four points.

The experiments and results per se are fine, but the context in which they are placed is highly misleading and this needs to be significantly modified.

Please see the revised introduction.

Reviewer #2 (Recommendations for the authors):[…]1) The authors claim that the constructs with stronger activation domains display higher bound fractions and longer residence times on chromatin. The fraction of bound molecules is measured by calculating the spectra of diffusion coefficients and by kinetic modelling of single molecule displacements, approaches that have been validated by multiple studies. The claim about residence times is instead based on Fluorescence Recovery after Photobleaching experiments (FRAP, Figure 2D, Figure S4B-C, Figure S6D), and this analysis is a bit more problematic. Indeed, as the authors correctly report, multiple phenomena and multiple models can explain FRAP recovery in presence of a mixture of diffusion and binding (Sprague, Biophys J., 2004, Mueller, Curr. Op. Cell Biol., 2010), and slower recovery curves can be caused by either slower dissociation (longer residence times), faster association, or slower diffusion – or a combination of these. Identifying whether longer binding or more frequent association to chromatin would be important, especially considering that recent SMF data is highlighting that also in mammalian cells on-rates are more relevant than off-rates in controlling occupancy of binding sites (Schaepe et al.,, BiorXiv, 2025). One possibility to address these issues would be to fit the FRAP data with the kinetic models that estimate binding rates (Mueller et al., Biophys J, 2008). An alternative solution could be to resort to slow frame-rates single molecule tracking to retrieve the distribution of binding times, and then estimate off-rates with the one of the approaches previously developed for this goal (Chen, Cell, 2013, Loffreda, Nat. Comm. 2017, Hansen, eLife, 2017, Nguyen, Mol Cell, 2021).

Thank you for suggesting this comprehensive FRAP analysis. We have replaced our original analysis with that of (Mueller et al., 2012) and tabulated fit parameters in Supplementary Table 1. We agree that differences in FRAP recovery curves can be due to differences in dissociation (k_off_), association (k_on_) or diffusion (D); these are the parameters that (Mueller et al., 2010) seek to fit. However, we believe our FRAP measurements are very “reaction-dominant” for two reasons:

1 We have conducted the gradient-smoothing test outlined in (Mueller et al., 2012; Methods: FRAP analysis-step 6) and included this in Figure 2—figure supplement 5. This tests for the role of diffusion during FRAP recovery, and we found no evidence that diffusion plays a role in our movies.

2. We also argue that diffusion has occurred before the first frame has been imaged after photobleaching. With the mobile diffusion coefficient we measured from SMT (slower population: D=2 µm^2^/s) and the radius of our bleach spot (0.75 µm), we can solve the equation:

mean squared displacement = 4 * D * t for t and calculate that, on average, molecules would diffuse from the edge of the FRAP spot to its center in ~70 milliseconds. Our frame rates were ~250 ms. We have also included this argument in Methods: FRAP analysis-step 6.

Given that diffusion is much faster than the observed recovery time, we can conclude that the measured rate is approximately equal to the rate of dissociation (which is equal to the rate of association in this steady-state system).

2) The discussion of the results on the p300 inhibitor (Figure 5A, incorrectly mislabeled as Figure 5B in the figure legend) was not very clear for me. The authors show that a drug that stabilize p300 binding on chromatin, also stabilize the binding of a sTF. Yet, this could be an indirect effect, and does not directly demonstrate that TFs can be tethered to chromatin through interactions with p300. At a minimum the effects of the p300 inhibitor on the chromatin interactions of same DBD, but without an activator domain, should be shown to support the authors' claim. A similar control would help discussing the results on the BAF inhibitor (Figure 5B, incorrectly mislabeled as Figure 4A in the figure legend).

We apologize for the errors. The logic of the p300 inhibitor experiment is confusing, so we have revised this section of the manuscript.

First, we have added a figure showing the effects of the drugs on a synthetic TF without any activation domain (Figure 6).

Second, we have added a new pharmacological perturbation: norstictic acid, which selectively inhibits VP16 interactions with Med25 (pg 17 line 395). This perturbation more directly supports the idea that activation domain coactivator interactions tether the TF to chromatin. It is still not definitive evidence.

Garlick JM, Sturlis SM, Bruno PA, Yates JA, Peiffer AL, Liu Y, Goo L, Bao L, De Salle SN, Tamayo-Castillo G, Brooks CL 3rd, Merajver SD, Mapp AK. 2021. Norstictic acid is a selective allosteric transcriptional regulator. *J Am Chem Soc* 143:9297–9302.

Finally, we have added a proximity-assisted photoactivation (PAPA) experiment that shows the HIF1α activation domain is in close proximity to p300 (Figure 5). Furthermore, the two alleles that increased HIF1α strength and chromatin association have increased proximity to p300. Molecules in proximity to p300 are also more likely to be bound to chromatin. This more direct measurement further supports the idea that activation domain interactions with coactivators can tether TFs to chromatin. This experiment demonstrates how PAPA has been applied to measure TF-coactivator interactions in vivo, which we anticipate will be a powerful new experimental approach.

3) This work clearly demonstrates that stronger TADs result in stronger/more frequent TF binding 'genome-wide'. However the data presented in this manuscript does not allow to conclude if such stronger binding has some specific effects at the binding sites, or if stronger regulation and stronger genomewide binding are two independent events. Performing chromatin immunoprecipitation(or single molecule footprinting, as in Doughty et al., 2024) experiments at the binding sites of the reporter gene, to test the occupancy of the same DBD with different TADs would significantly increase the impact of this work.

As mentioned above, we have included CUT&RUN experiments for six synthetic TFs (Figure 7).

4) The authors should cite some previous papers that, by using SMT, have tested the effect of removing TADs on endogeneous TF binding and diffusion. Some of them (Chen et al., Cell 2014) have shown that less binding to chromatin is observed, in agreement with this study, while others (Callegari et al., Embo Rep, 2019, Mazzocca et al., Nat Comm., 2023), have shown no effect on binding, but a modulation of their diffusion properties.

Thank you for these suggestions. Please see the revised introduction (pg 3 line 92-93).

5) The authors present an interesting methodological approach to extract errors from SMT data, using cell-based boot-strapping. This allows to provide estimates of Bound fractions with error bars (Figure 3D). The authors should include error estimates using the same approach also in Figure 2C.

We have added error bars to Figure 2C.

[Editors’ note: what follows is the authors’ response to the second round of review.]

Please address the concerns of reviewer 1 that are listed below.Reviewer #1 (Recommendations for the authors):Although the revised paper is improved by better literature citation and a new experiment, I'm sorry to say that my overall opinion hasn't changed. As I stated before, the experiments are useful, but the main conclusions are incremental, entirely expected, and hardly challenge traditional views of anything. I don't have a problem with the paper being published in eLife, but I recommend rejection if the authors refused to remove all statements claiming that their paper challenges traditional/previous views.1. As can be seen from the figure and associated text in their response letter, the authors have misunderstood my main point. This authors' figure concerns cooperative effects involving separate DNA-binding proteins bound to separate sequences. That can certainly happen, but it has nothing to do with my main point, and it has nothing to do with their observations. Instead, my point is that activation domains directly interact with chromatin-modifying activities (e.g. nucleosome remodelers such as Swi/Snf and histone acetylases such as SAGA/Gcn5) that directly interact with DNA and/or histones. As such, the cooperativity that causes increased chromatin association does not involve a second DNA-binding but rather occurs between a single DNA-binding protein and the recruited chromatin-modifying activity. All DNA-binding proteins have low-level affinity for all sequences (typically called non-specific binding), and the interaction of activation domains and chromatin-modifying activities is independent of specific DNA-binding and would occur throughout the genome, albeit at much lower levels for a non-specific sequence. Thus, it is obvious that the combination of specific DNA-binding domains and associated chromatin-modifying activities will result in higher genome association and residence times. The paper is useful in that it provides experimental evidence and measurements for this, but it doesn't break any new ground, and it certainly doesn't challenge traditional views.

We apologize that we did not understand your original comments. Thank you for clarifying.

If we understand correctly, you are saying: sequence specific TFs bind the genome and recruit chromatin modifying complexes. Once they are recruited, the chromatin modifying complexes bind to nucleosomes or DNA, but without any sequence specificity. The molecular bridging and avidity of a TF bound to both DNA and a chromatin modifying complex bound to a nucleosome increase TF dwell time on DNA (Author response image 1,Panel B and D). This recruitment can occur at specific binding sites (C+D) or at non-specific DNA (A+B). We think you are saying that B will have a longer dwell time than A, and D will have a longer dwell time than C.

If this is your model, we agree that it is compatible with our data. We have added a sentence to the discussion to include this interpretation:“We cannot distinguish whether the coactivator directly binds DNA, binds a nucleosome, or binds another TF bound to DNA, but we suspect all three binding modes are involved.” (pg 22 line 520)

However we do not think it was “obvious” that this was the only possible model *a priori*. The nucleus is chalk full of TF-TF interactions (Abidi et al. 2025) and TF-RNA interactions (Nabeel-Shah et al. 2024). We focused on AD:coactivator interactions through drug perturbations and PAPA experiments.

2. Although the citations are significantly improved, the paper still hypes the results by claiming to overturn the "traditional view". No one has ever claimed the so-called traditional view that "the DBD is solely responsible for binding chromatin". As such, the results here cannot challenge a view that no one has expressed or even believed. Instead, the authors have invented this "traditional view" and used it as a straw man to shoot down. All the early work on activation domains and the vast majority of subsequent work ignored generic chromatin association because it was hard to imagine how it would contribute to the specificity of gene regulation. Even with this paper, the biological significance of generic chromatin association is unclear. I do agree that the measurements here are useful, but this is far from overturning a traditional view that has not even been expressed.

We have revised the introduction to remove the rhetoric about the DBD being “solely responsible for binding chromatin.” In the abstract, we now say the DBD is “primarily responsible for binding DNA and chromatin.” Last week the corresponding author attended a conference entitled *“Rules of protein-DNA Interactions,”* where he polled the audience of ~36 PIs about this word choice. This group felt the revised sentence is much more appropriate than our original sentence. The revised sentence is:

Despite evidence to the contrary, DBDs are often assumed to be the primary mediators of TF interactions with DNA and chromatin. (pg 1 line 15)

We instituted a major change in how we motivate the manuscript. We now motivate the work by setting out to measure the relative contribution of DBDs and activation domains to chromatin binding. We believe that the relative contributions cannot be predicted and need to be measured.

Revised abstract (with emphasis added to major changes):

“Transcription factors regulate gene expression with DNA-binding domains (DBDs) and activation domains. Despite evidence to the contrary, DBDs are often assumed to be the primary mediators of TF interactions with DNA and chromatin. Here, we used fast single-molecule tracking of transcription factors in living cells to show that short activation domains can control the fraction of molecules bound to chromatin. Stronger activation domains have higher bound fractions and longer residence times on chromatin. Furthermore, mutations that increase activation domain strength also increase chromatin binding. This trend was consistent in four different activation domains and their mutants. This effect further held for activation domains appended to three different structural classes of DBDs. Stronger activation domains with high chromatin-bound fractions also exhibited increased binding to the p300 coactivator in proximity-assisted photoactivation experiments. Genome-wide measurements indicate these activation domains primarily control the occupancy of binding rather than the genomic location. Taken together, these results demonstrate that very short activation domains play a major role in tethering transcription factors to chromatin.”

We also removed the paragraphs about the history of the recruitment model and TF modularity. Instead we include the following text:

“Both DBDs and activation domains contribute to DNA binding and chromatin binding. DNA binding results from direct protein-DNA interaction, is well established for DBDs in vitro, and is measured in vivo with single-molecule footprinting (Doughty et al., 2024; Stormo, 2013).

Chromatin binding includes DNA binding but can also result from protein-protein interactions with histones, other TFs, corepressors, or coactivators that can indirectly tether a TF to DNA (Spitz and Furlong, 2012). Nearly all cell-based assays of TF genome localization (e.g. ChIP-seq or CUT&RUN) measure the combination of DNA binding and chromatin binding. Some activation domains interfere with DNA binding (Golemis and Brent, 1992). in vitro, interactions between activation domains and DBDs can increase DNA binding specificity (Krois et al., 2018; Liu et al., 2008; Ning et al., 2022; Sun et al., 2021). IDRs can increase non-specific binding (Baughman et al., 2022), regulate DNA binding via post-translational modifications (Pufall et al., 2005), or allosterically regulate DNA binding (Li et al., 10 2017). There is long-standing evidence that the DBD does not confer all genomic targeting information because in ChIP-seq experiments, 30-70% of peaks lack a motif for the query TF (Harrison et al., 2011; Jana et al., 2021; Kvon et al., 2012; Spitz and Furlong, 2012; Teytelman et al., 2013). There is also low agreement between the promoters a TF binds and the genes it regulates (Mahendrawada et al., 2025). Finally, there is extensive literature demonstrating how long IDRs—and not DBDs—control TF genomic localization (Brodsky et al., 2020; Gera et al., 2022; Hurieva et al., 2024; Jonas et al., 2023; Kumar et al., 2023; Mindel et al., 2024a, 2024b). These studies show that long IDRs are necessary and sufficient to localize TFs to target promoter genes; minimal activation domains were reported not to contribute (Brodsky et al., 2020). Although the original data could not distinguish between direct DNA binding and chromatin binding via protein-protein interactions with DNA-bound factors (reviewed in (Staller, 2022)), recent studies suggest an IDR can bind DNA directly (Strugo et al., 2025). This genomic targeting by IDRs has also been seen in human cells (Goldman et al., 2023). These findings can explain why removing the DBD can increase the number of genomic binding sites detected by ChIP-seq (Chen et al., 2014), have little effect on genome binding (Cowling and Cole, 2007), or complement null mutants (Copeland et al., 1996). We have shown that long IDRs control the fraction of TF molecules bound to chromatin for factors in the HIF family (Chen et al., 2022). Other single-molecule imaging studies reported that IDRs impact nuclear diffusion but not genome binding (Callegari et al., 01 2019; Mazzocca et al., 2023). A single-molecule footprinting assay found that a strong activation domain can increase TF occupancy (Doughty et al., 2024). Together, these studies demonstrate how long IDRs or activation domains modulate chromatin binding.

In this study, we asked whether short activation domains within an IDR are sufficient to control chromatin association. While both DBDs and activation domains contribute to DNA binding and chromatin binding, the relative magnitude of these contributions cannot be predicted and must be measured. Activation domains recruit remodelers, like BAF, which move nucleosomes (Kadoch and Crabtree, 2015), coactivators, like p300/CBP, which modify chromatin (DelRosso et al., 2024; Dyson and Wright, 2016), and general transcription components like the Mediator complex (Currie et al., 2017; Henley et al., 2020; Lauberth et al., 02 2013; Malik and Roeder, 2010; Tang et al., 07 2013) that recruits RNA polymerase II (Alerasool et al., 2022; Gill et al., 1994). Although mechanistic understanding of activation domains still lags far behind DBDs, recent screens have cataloged many activation domains (DelRosso et al., 2023; Erijman et al., 2020; Morffy et al., 2024; Sanborn et al., 2021) and shown how the strength of acidic activation domains, the oldest and largest class (Sigler, 05 1988), depends on a balance of acidic, aromatic, and leucine residues (DelRosso et al., 2023; Staller et al., 2022, 2018; Udupa et al., 2024). For consistency, we define activation domain strength as the fluorescence of a GFP reporter construct driven by a heterologous promoter.”

In addition, we made minor changes to two paragraphs in the discussion:

“How a TF binds the genome is controlled by both DBDs and IDRs, but it remains difficult to predict the relative contributions of these two regions. We demonstrated that very short activation domains can control the fraction of TF molecules bound to chromatin, sometimes accounting for more binding than the DBD. Mutations that increase activation domain strength in reporter assays increase the fraction of molecules bound to chromatin. Conversely, mutations that decrease activation domain strength decrease the fraction of molecules bound to chromatin. This trend holds for allelic series of four acidic activation domains, for three structurally diverse DBDs, and for synthetic and full-length TFs.”

3. Regarding the key concept of modularity of DNA-binding and activation domains, the authors have invented ideas that were not expressed or even contemplated in previous papers. The original papers on modularity were concerned with specific DNA-binding and transcriptional activation. They were published well before the concept of co-activators, and indeed the prevailing view based on bacterial activators is that activation domains would contact some component of the basic Pol II machinery. Indeed, there were numerous experiments claiming that various general factors (e.g. TBP, TFIIB, TFIIA) were the target. These early papers establishing modularity were also well before any connection to chromatin, and at the time, chromatin was largely ignored in the gene expression field. Eventually, it became clear that activation domains not only stimulated transcription directly (mostly through Mediator) but also indirectly through chromatin via Swi/Snf and SAGA. The connection of activation domains and chromatin is 30 years old, so using early papers on modularity to claim that activation domains had nothing to do with chromatin is historically unfair. And, this 30-year connection between activation domains and chromatin-modifying activities is what renders the results in this paper entirely predictable.

We have removed the rhetoric about transcription factor modularity (see above.)

We were genuinely surprised by our result. We expected chromatin binding to be controlled by the long, intrinsically disordered regions between DBDs and activation domains, as shown by (Brodsky et al. 2020; Chen et al. 2022). Brodsky showed for Msn2 that the activation domains were not responsible for genome localization (Brodsky et al. 2020).

We think the main value of our paper is quantitatively measuring relative contributions of the DBD and very short activation domains on chromatin binding under physiological conditions in live cells. The magnitude of the activation domain contribution could not be predicted *a priori*.

Reviewer #2 (Recommendations for the authors):The authors have satisfactorily addressed my comments.In particular, the addition of CUT&RUN experiments clarifies that swapping ADs only slightly impacts where the TF will bind, but rather modulates binding frequency and/or stability at pre-determined sites. I think that this result should be highlighted at the end of the introduction. Also, as a minor point I would be curious to see if the motifs bound by the different TFs variants are identical, an observation that could further support their observation.

Thank you for these comments and suggestions. We have looked for motifs in the CUT&RUN data with MEME using scrambled sequences as the controls. We did not find anything that jumped out at us. All the TFs share runs of A’s, which could be nucleosome repelling sequences.